# ENHANCING LARGE LANGUAGE MODEL REASONING VIA SELECTIVE CRITICAL TOKEN FINE-TUNING

## ABSTRACT

Large language models (LLMs) primarily rely on supervised fine-tuning (SFT) as a key method to adapt pre-trained models to domain-specific tasks such as mathematical reasoning. However, standard SFT uniformly penalizes all tokens, neglecting that only a small subset of critical tokens determines reasoning correctness. This uniform supervision often causes reduced output diversity and limited generalization. We propose **Critical Token Fine-tuning (CFT)**, a simple yet effective approach that updates only tokens identified as functionally indispensable via counterfactual perturbations. By focusing gradient signals on these decisive reasoning steps while preserving the diversity of non-critical tokens, CFT can enhance both generation and diversity. Extensive experiments on five models across three families (Qwen, OLMo, LLaMA) and eleven mathematical reasoning benchmarks show that CFT, despite fine-tuning on less than 12% of tokens, consistently outperforms standard SFT. Moreover, CFT enables test-time scaling through improved sampling diversity and provides a stronger initialization for reinforcement learning, sustaining performance gains in later training stages while maintaining higher entropy for better exploration. These results highlight CFT as a practical and general framework for efficient and robust LLM fine-tuning.

## 1 INTRODUCTION

Large language models (LLMs) have achieved remarkable progress across a wide range of complex tasks, driven by the rapid scaling of both model parameters and training data (Fedus et al., 2022; Achiam et al., 2023; AI@Meta, 2024; Team, 2024; Brown et al., 2020). To adapt these general-purpose models to specialized downstream tasks (Yu et al., 2024), the prevailing paradigm is supervised fine-tuning (SFT) (Sanh et al.; Ruan et al., 2025), which optimizes on labeled prompt–response pairs using a maximum likelihood objective (Ouyang et al., 2022). SFT can also serve as an initialization for reinforcement learning (RL), providing a strong starting point that aids further RL optimization (Chu et al., 2025; Li et al., 2025).

The effectiveness of SFT is highly dependent on data quality (Zhou et al., 2023; Li et al., 2024). In mathematical reasoning, a common practice is to filter data by verifying the correctness of final answers (He et al., 2025; Yu et al., 2025). Nevertheless, correctness at the response level does not guarantee that every token in the reasoning chain is an "ideal" choice within its context. Even correct solutions may contain tokens with low reward (Liu et al., 2025). Blindly imitating all tokens in such imperfect reasoning traces reduces optimization efficiency reward (Liu et al., 2025). Moreover, only a small subset of tokens truly determines the correctness of the final answer. Uniformly penalizing all deviations risks suppressing valid alternative reasoning paths and unintentionally eroding the pre-trained model's diversity (Li et al., 2025; Kim et al.; O'Mahony et al., 2024).

Recent studies have begun addressing this issue by assigning non-uniform importance to tokens. For example, TIS-DPO (Liu et al., 2025) estimates token weights using the logit differences between a reward and penalty model, while other approaches rely on model confidence scores (Wu et al., 2025a). However, these methods either require two additional training models or fail to account for the actual functional role of tokens in the reasoning chain. As a result, a direct way to identify which tokens are truly decisive for final correctness remains missing.

To bridge this gap, we introduce Critical Token Fine-tuning (CFT), a novel approach that refines SFT by applying gradient updates only to tokens proven indispensable for reasoning correctness.

We identify these critical tokens through a counterfactual perturbation process: for each token in a correct solution, we replace it with alternative candidates and regenerate the continuation. If all perturbations yield incorrect final answers, the original token is marked as critical, reflecting its functional necessity. To accelerate this process, we leverage parallel decoding of alternative paths for a given response, achieving a speedup of over 25x on Qwen2.5-7B (Team, 2024).

By focusing optimization on decisive reasoning steps and excluding non-critical tokens, CFT outperforms standard SFT in both accuracy and diversity. This approach also serves as an **effective initialization for RL**, enabling sustained performance gains and better exploration. Furthermore, CFT has shown broad applicability, as demonstrated by its success in medical QA (Jin et al., 2021), proving its effectiveness beyond mathematical reasoning. Our main contributions are threefold:

- We propose a practical, model-agnostic framework for identifying critical tokens via counterfactual perturbations and demonstrate that fine-tuning on only a small fraction of critical tokens (often $< 12\%$) can surpass standard SFT.
- We show that CFT enables test-time scaling by preserving output diversity, improving inference-time sampling (`Pass@N`) with a broader set of high-quality solutions.
- When used as initialization for RL, CFT-trained models outperform standard SFT models, achieving sustained performance gains and maintaining higher entropy during RL optimization, which enhances exploration.

## 2 RELATED WORK

**Token-level supervision**  Supervised Fine-Tuning (SFT) is the standard paradigm for adapting pre-trained LLMs to downstream tasks by training on labeled examples with a maximum likelihood objective (Wei et al., 2022; Ouyang et al., 2022). While effective, recent work has highlighted that SFT's uniform treatment of all tokens can suppress valid reasoning alternatives and reduce output diversity(Li et al., 2025; Kim et al.; O'Mahony et al., 2024). To address these issues, recent research explores assigning different importance to tokens. For instance, DFT stabilizes gradient updates for each token by dynamically rescaling the objective function with the probability of this token (Wu et al., 2025a). Other methods, such as TIS-DPO and cDPO (Lin et al., 2025; Liu et al., 2025), estimate token importance through logits comparison but require training two additional models. RHO-1 (Lin et al., 2024) shows that only a fraction of tokens in corpora drive substantial loss reduction in the pre-training stage. These approaches highlight the need for efficient methods to identify truly decisive tokens for reasoning.

**Rollout-based verification**  Using rollouts to improve model performance is a common strategy. Some methods apply them at inference time, such as self-consistency and search-based approaches (Wang et al., 2023; Yao et al., 2023; Hao et al., 2023; Besta et al., 2024), which aggregate or explore multiple reasoning paths. Others employ rollouts for verification, e.g., Math-Shepherd checks intermediate steps by sampling continuations (Wang et al., 2024). More fine-grained approaches, such as cDPO (Liu et al., 2025), estimate token importance by perturbing positions and running 64 rollouts per token. While effective, this procedure is computationally costly and relies on statistical estimation rather than direct causal analysis. In contrast, our method uses counterfactual substitution with only one or two greedy rollouts per token, enabling efficient evaluation without auxiliary models.

## 3 METHOD

### 3.1 PRELIMINARIES

Traditional supervised fine-tuning (SFT) minimizes the cross-entropy loss over all tokens in the training set:

$$\mathcal{L}_{\text{SFT}}(\theta) = - \sum_{(Q,Y) \in \mathcal{D}} \sum_{t=1}^{T} \log P(y_t \mid Q, y_{<t}; \theta), \tag{1}$$

where $(Q, Y)$ is a question–response pair from $\mathcal{D}$ and $y_t$ is the $t$-th token in $Y$.

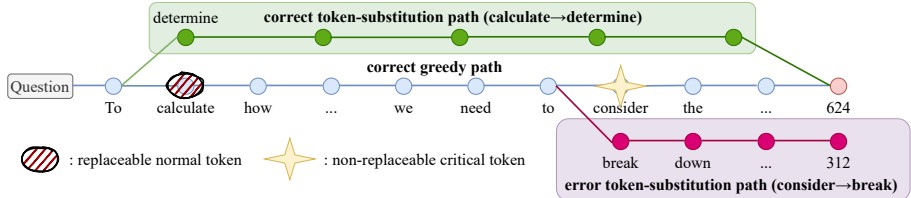

Figure 1: Identifying critical tokens in CFT via counterfactual perturbation. A token is deemed non-critical if substituting it maintains correctness, indicating it is replaceable (e.g., `calculate` → `determine`, green path). It is critical if if the substitution causes an incorrect answer (e.g., `consider` → `break`, red path).

Let $z_{t,v}$ denote the logit for vocabulary item $v \in \mathcal{V}$ at position $t$, and define $p_{t,v} = \text{softmax}(z_t)_v = \frac{e^{z_{t,v}}}{\sum_{u \in \mathcal{V}} e^{z_{t,u}}}$. For the gold token $g_t \equiv y_t$, the per-token loss is $\ell_t = -\log p_{t,g_t}$ with gradient:

$$\frac{\partial \ell_t}{\partial z_{t,v}} = p_{t,v} - \mathbb{I}[v = g_t]. \tag{2}$$

Aggregating over the dataset gives $\frac{\partial \mathcal{L}_{\text{SFT}}}{\partial z_{t,v}} = \sum_{(Q,Y)} \left( p_{t,v} - \mathbb{I}[v = g_t] \right)$, indicating that all tokens are weighted equally. Equation 2 enforces $p_{t,g_t} \to 1$ and $p_{t,v \neq g_t} \to 0$, i.e., a mode-seeking update that collapses the token distribution at every position. This uniform treatment is problematic for reasoning tasks, where only a small fraction of tokens are truly decisive for correctness.

## 3.2 IDENTIFYING CRITICAL TOKENS

Given a pre-trained model $M_\theta$ and training dataset $\mathcal{D} = \{(Q_i, A_i)\}$, we first construct a subset $\mathcal{D}_{\text{correct}}$ from instances where the model, under greedy decoding, produces correct reasoning (Wu et al., 2025b). For each question $Q_i$, we generate

$$Y_i = (y_1, y_2, \ldots, y_T), \quad y_t = \arg\max_{y \in \mathcal{V}} P(y \mid Q_i, y_{<t}; \theta), \tag{3}$$

and retain $(Q_i, Y_i)$ only if the final answer extracted from $Y_i$ matches $A_i$:

$$\mathcal{D}_{\text{correct}} = \{(Q_i, Y_i) : \mathcal{F}(Y_i, A_i) = 1\}, \tag{4}$$

where $\mathcal{F}(\cdot, \cdot)$ is a verification function returning 1 for correctness.

For each $Y \in \mathcal{D}_{\text{correct}}$, we assess the criticality of token $y_t$ via counterfactual perturbation, as shown in Figure 1. Let $\{y_t^{(2)}, y_t^{(3)}, \ldots, y_t^{(k)}\}$ denote the 2-nd through $k$-th most probable alternatives. For each $y_t^{(j)}$, we form the counterfactual continuation:

$$\tilde{Y}_t^{(j)} = (y_1, \ldots, y_{t-1}, y_t^{(j)}, \tilde{y}_{t+1}, \ldots, \tilde{y}_{T'}), \tag{5}$$

where the suffix $(\tilde{y}_{t+1}, \ldots)$ is generated greedily. Token $y_t$ is marked **critical** if all reasonable alternatives fail to preserve correctness:

$$c_t = \mathbb{I}\left[ \bigwedge_{j=2}^{k} \mathcal{F}(\tilde{Y}_t^{(j)}, A) = 0 \right]. \tag{6}$$

**Parallel Critical Identification.** To reduce computational time, we evaluate all positions in parallel for a fixed $j$:

$$\text{Batch}_j = \{[Q, y_1^{(j)}], [Q, y_1, y_2^{(j)}], \ldots, [Q, y_1, \ldots, y_{T-1}, y_T^{(j)}]\}, \tag{7}$$

allowing efficient amortization of counterfactual rollouts.

### 3.3 SELECTIVE FINE-TUNING OBJECTIVE

Fine-tuning is then restricted to critical positions:

$$\mathcal{L}_{\text{CFT}}(\theta) = -\frac{\sum_{(Q,Y)\in\mathcal{D}_{\text{correct}}} \sum_{t=1}^{T} c_t \log P(y_t \mid Q, y_{<t}; \theta)}{\sum_{(Q,Y)\in\mathcal{D}_{\text{correct}}} \sum_{t=1}^{T} c_t}. \tag{8}$$

This objective removes training pressure from replaceable tokens. While SFT sharpens the distribution at every step, CFT preserves entropy at non-critical positions and focuses optimization only on indispensable decisions.

The gradient of the per-token CFT loss, $\ell_t^{\text{CFT}} = -\frac{c_t}{Z} \log p_{t,g_t}$ with $Z = \sum_{(Q,Y)\in\mathcal{D}_{\text{correct}}} \sum_t c_t$, is

$$\frac{\partial \ell_t^{\text{CFT}}}{\partial z_{t,v}} = \frac{c_t}{Z} \left( p_{t,v} - \mathbb{I}[v = g_t] \right). \tag{9}$$

When $c_t = 0$, the gradient is zero; when $c_t = 1$, it matches the SFT gradient up to a constant. Thus, CFT applies mode-seeking updates only at positions where alternatives provably fail.

Overall, CFT replaces uniform token weighting with a data-driven criticality mask, thereby avoiding distribution collapse on replaceable tokens while strengthening indispensable reasoning steps. This selective training leads to richer output distributions and improved generalization in reasoning tasks.

## 4 EXPERIMENTS

### 4.1 SETUP AND IMPLEMENTATION DETAILS

**Training Dataset and Models.** We use the GSM8K training set (Cobbe et al., 2021) as the primary corpus, which contains math word problems with step-by-step solutions. For each base model, including Qwen2.5-3B, Qwen2.5-7B (Team, 2024), Qwen3-8B (Team, 2025), LLaMA3.1-8B (AI@Meta, 2024), and OLMo2-7B (OLMo et al., 2024), we perform greedy decoding on GSM8K to construct model-specific subsets consisting of correctly solved questions (Wu et al., 2025b). These subsets are then annotated with critical tokens according to Equation 6 with $k = 3$. The number of correctly solved instances for each model, as well as the proportion of tokens identified as critical, are reported in Appendix B.1. In Section 5, we further extend the annotation procedure to include offline data from other models and resample on unsolved questions.

**Baselines.** We compare CFT against representative fine-tuning strategies:

• **Supervised Fine-Tuning (SFT)** uniformly updates all tokens with a cross-entropy loss.

• **Dynamic Fine-Tuning (DFT)** (Wu et al., 2025a) rescales token-wise losses according to predicted probabilities to stabilize optimization.

• **Entropy-based Selection** (Wang et al., 2025) marks high-entropy tokens as critical tokens.

• **Attention-based Selection** (Vaswani et al., 2017) labels tokens with larger attention as critical.

All methods are instantiated with full fine-tuning in the main experiments and LoRA in Table 6 of the Appendix for completeness. For Entropy and Attention baselines, the fraction of tokens selected is matched to that of CFT, enabling a direct comparison of the effectiveness of the critical tokens.

**Benchmarks.** We evaluate on eleven mathematical reasoning benchmarks spanning diverse difficulty levels, including GSM8K (Cobbe et al., 2021), MATH (Hendrycks et al.), SVAMP (Patel et al., 2021), ASDiv (Miao et al., 2020), MAWPS (Koncel-Kedziorski et al., 2016), CARP_En (Zhang et al., 2023a), TabMWP (Lu et al.), Minerva_Math (Lewkowycz et al., 2022), Gaokao2023En (Zhang et al., 2023b), OlympiadBench (He et al., 2024), and College_Math (Tang et al., 2024). Our evaluation pipeline follows the official Qwen2.5-Math framework[1].

---

[1] https://github.com/QwenLM/Qwen2.5-Math

Table 1: Comparison of fine-tuning methods on eleven mathematical reasoning benchmarks across five LLMs. **Attn** denotes the Attention-based Selection baseline. **Avg.** denotes the average accuracy across all 11 benchmarks.

| Methods | Avg. | GSM8K | Math | SVAMP | ASDiv | MAWPS | CARP | TabMWP | Minerva | Gaokao | Olympiad | College |
|---|---|---|---|---|---|---|---|---|---|---|---|---|
| **Qwen2.5-3B** | 43.3 | 65.5 | 34.6 | 80.6 | 74.9 | 81.8 | 38.5 | 37.7 | 12.1 | 29.9 | 10.4 | 10.7 |
| SFT | 49.3 | 69.8 | 43.2 | 81.2 | 78.5 | 85.2 | 44.0 | 46.2 | 15.8 | 39.0 | 18.1 | 20.8 |
| DFT | 51.1 (+1.8) | 71.9 | 45.0 | 84.5 | 80.8 | 87.1 | 45.2 | 50.7 | 18.4 | 47.4 | 17.9 | 23.5 |
| Entropy | 50.1 (+0.8) | 68.9 | 45.3 | 82.9 | 79.0 | 85.1 | 45.1 | 49.2 | 16.5 | 38.4 | 18.7 | 22.5 |
| Attn | 51.5 (+2.2) | 69.4 | 47.5 | 82.3 | 78.1 | 86.1 | 50.0 | 48.4 | 18.4 | 42.1 | 20.7 | 23.6 |
| CFT | **55.1 (+5.8)** | 71.9 | 52.5 | 81.2 | 83.1 | 90.9 | 53.6 | 54.6 | 20.2 | 45.2 | 19.9 | 33.3 |
| **Qwen2.5-7B** | 48.3 | 75.2 | 39.6 | 85.5 | 78.7 | 86.4 | 41.4 | 46.0 | 15.4 | 35.8 | 14.8 | 12.0 |
| SFT | 52.8 | 77.0 | 47.3 | 88.0 | 82.4 | 89.7 | 46.2 | 46.3 | 22.4 | 44.2 | 21.6 | 15.4 |
| DFT | 53.5 (+0.7) | 78.5 | 49.6 | 87.8 | 81.9 | 88.6 | 47.0 | 51.1 | 20.2 | 42.3 | 23.0 | 18.3 |
| Entropy | 55.4 (+2.6) | 78.6 | 53.9 | 87.9 | 83.4 | 91.7 | 49.2 | 48.6 | 25.4 | 43.9 | 25.3 | 21.6 |
| Attn | 55.4 (+2.6) | 78.8 | 53.9 | 87.9 | 83.7 | 91.6 | 49.2 | 48.4 | 25.4 | 43.9 | 25.5 | 21.5 |
| CFT | **59.2 (+6.4)** | 81.5 | 59.8 | 90.2 | 85.8 | 92.6 | 53.9 | 53.1 | 26.8 | 48.6 | 28.4 | 30.0 |
| **Qwen3-8B** | 60.0 | 84.2 | 66.8 | 90.9 | 83.4 | 89.4 | 54.3 | 46.8 | 28.3 | 52.5 | 36.4 | 26.8 |
| SFT | 61.1 | 83.8 | 69.2 | 90.6 | 84.9 | 91.0 | 54.4 | 45.3 | 31.6 | 56.1 | 36.6 | 28.1 |
| DFT | 61.0 (-0.1) | 84.5 | 67.8 | 90.7 | 84.2 | 91.0 | 54.6 | 45.5 | 32.0 | 56.6 | 36.7 | 27.5 |
| Entropy | 60.7 (-0.4) | 84.3 | 67.4 | 91.3 | 84.9 | 91.8 | 54.0 | 46.3 | 29.8 | 54.3 | 36.6 | 26.5 |
| Attn | 61.1 (+0.0) | 85.7 | 68.9 | 91.2 | 85.0 | 91.5 | 54.4 | 44.3 | 29.8 | 56.4 | 36.3 | 28.1 |
| CFT | **63.5 (+2.4)** | 88.0 | 70.7 | 93.5 | 87.9 | 93.8 | 56.1 | 52.6 | 32.7 | 58.2 | 37.2 | 28.0 |
| **LLaMA3.1-8B** | 27.8 | 23.4 | 12.9 | 60.7 | 56.2 | 65.1 | 13.9 | 42.4 | 9.2 | 13.2 | 3.0 | 6.0 |
| SFT | 38.7 | 57.2 | 19.9 | 70.4 | 74.4 | 90.0 | 24.6 | 49.2 | 7.0 | 18.7 | 4.4 | 9.4 |
| DFT | 38.6 (-0.1) | 57.8 | 19.0 | 71.8 | 74.2 | 91.3 | 21.3 | 48.7 | 8.5 | 18.7 | 4.3 | 8.7 |
| Entropy | 37.8 (-0.9) | 52.1 | 18.4 | 70.6 | 71.5 | 89.4 | 21.6 | 51.1 | 9.2 | 17.7 | 5.0 | 9.7 |
| Attn | 36.7 (-2.0) | 60.0 | 16.0 | 66.2 | 72.1 | 87.6 | 20.9 | 48.6 | 7.0 | 15.1 | 4.0 | 6.6 |
| CFT | **39.2 (+0.5)** | 55.6 | 18.9 | 70.1 | 72.7 | 87.4 | 21.9 | 57.2 | 8.5 | 19.7 | 4.7 | 14.1 |
| **OLMo2-7B** | 33.2 | 58.2 | 13.3 | 71.3 | 64.6 | 82.3 | 12.5 | 31.3 | 5.5 | 16.1 | 3.9 | 6.1 |
| SFT | 39.7 | 69.1 | 20.9 | 77.0 | 72.8 | 84.4 | 24.1 | 48.2 | 4.8 | 17.9 | 5.2 | 12.6 |
| DFT | 41.8 (+2.1) | 71.3 | 22.2 | 75.5 | 75.9 | 86.2 | 23.2 | 61.4 | 5.9 | 20.3 | 4.1 | 14.2 |
| Entropy | 39.3 (-0.4) | 68.3 | 19.3 | 75.3 | 71.7 | 88.9 | 19.1 | 53.4 | 3.7 | 19.7 | 3.1 | 9.3 |
| Attn | 38.6 (-1.1) | 70.0 | 16.4 | 77.5 | 76.9 | 94.4 | 19.6 | 42.7 | 4.8 | 12.5 | 3.0 | 6.2 |
| CFT | **42.3 (+2.6)** | 74.3 | 21.0 | 75.2 | 78.7 | 92.0 | 20.3 | 61.2 | 4.8 | 21.6 | 4.3 | 12.1 |

**Training Details.** All fine-tuning experiments are conducted with the OpenRLHF framework[2]. Following prior observations that different adaptation schemes favor different learning rates (Biderman et al., 2024), we adopt smaller rates for full fine-tuning and larger ones for LoRA. To ensure fairness, we perform a hyperparameter sweep rather than fixing a single configuration. For full fine-tuning, we vary batch sizes $\{16, 32, 128\}$ and learning rates $\{2e\text{-}6, 5e\text{-}6, 2e\text{-}5\}$. For LoRA (reported in Table 6), we use higher learning rates $\{2e\text{-}5, 5e\text{-}5, 2e\text{-}4\}$ with the same batch size grid. All models are trained for three epochs using the Adam optimizer with a cosine decay schedule, a 3% warmup ratio, and BF16 precision. Unless otherwise noted, results in the main tables correspond to the best-performing hyperparameter configuration for each method. All experiments are conducted on 8 NVIDIA A100-80G GPUs.

## 4.2 Main Results

Table 1 summarizes the performance of different fine-tuning strategies across eleven reasoning benchmarks. Several consistent findings emerge from the results.

**Selective updates on critical tokens yield consistent gains.** Across all five models, CFT consistently surpasses all baselines. This confirms that counterfactual perturbation identifies decisive reasoning steps more reliably than heuristic criteria such as entropy or attention. We further note that DFT performs on par with SFT, likely because model-generated responses assign high confidence to most tokens, leaving little room for probability-based reweighting to have an effect.

**Strong generalization beyond the training domain.** Although the fine-tuning corpus is derived solely from GSM8K, the improvements extend broadly to out-of-domain benchmarks, including arithmetic reasoning (e.g., SVAMP, ASDiv), multi-step word problems (e.g., MAWPS, TabMWP, CARP), and advanced mathematics (e.g., Math, Minerva, Olympiad). This indicates that **focusing gradient updates on critical reasoning steps promotes compositional generalization** rather than overfitting to the training distribution.

**Robust improvements across model families.** Performance gains are observed across all backbone architectures, spanning Qwen, LLaMA, and OLMo. This demonstrates that CFT is model-agnostic

---

[2]https://github.com/OpenRLHF/OpenRLHF

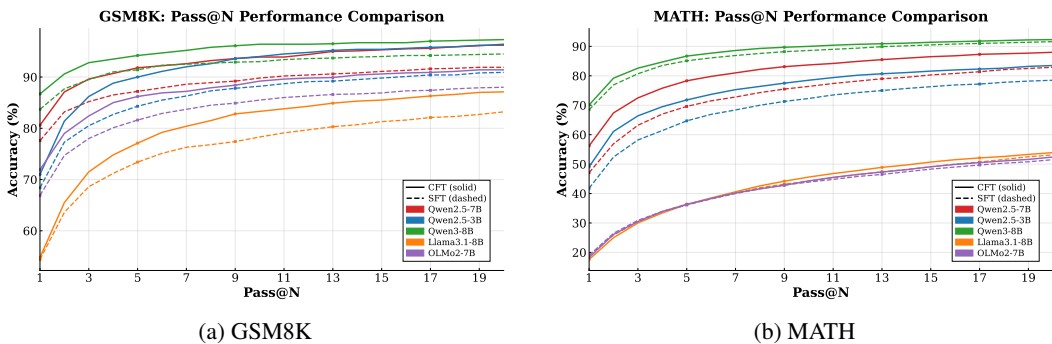

(a) GSM8K                                             (b) MATH

Figure 2: Pass@N comparison between CFT (solid lines) and SFT (dashed lines) across multiple backbones. Note that the vertical axis range is narrower in (a) than in (b).

and can be seamlessly applied to diverse architectures, providing a simple yet general recipe for enhancing reasoning ability in LLMs.

Overall, CFT delivers robust improvements over all baselines, across datasets and model families, underscoring its effectiveness as a general strategy for reasoning-oriented model adaptation.

### 4.3 INFERENCE-TIME SCALING

Inference-time strategies such as Best-of-$N$ (BoN) sampling (Charniak & Johnson, 2005; Stiennon et al., 2020), which generate multiple candidate outputs for each query, are widely adopted to enhance LLM reasoning (Welleck et al., 2024; Snell et al., 2025). The effectiveness of these methods relies on whether the generated candidates can sufficiently explore the solution space.

To quantify this effect, we adopt the **Pass@$N$** metric, which can be regarded as a special case of BoN (Chen et al., 2021). For a question $q$, let $\{o_i\}_{i=1}^N$ denote $N$ outputs and $\mathcal{F}(o_i)$ a verifier returning 1 if $o_i$ is correct and 0 otherwise. Then,

$$\text{Pass@}N(q) = \mathbb{I}\Big[\exists\, i \in \{1, \ldots, N\} \text{ s.t. } \mathcal{F}(o_i) = 1\Big]. \tag{10}$$

This metric evaluates whether at least one correct solution is found among $N$ attempts.

**Experimental setup.** We build upon the models fine-tuned in Section 4.2. To assess inference-time diversity, we evaluate Pass@$N$ performance on two benchmarks: GSM8K and MATH. For each query, we generate $N$ responses, with $N$ varying from 1 to 20, and compute Pass@$N$ accuracy using an automated verifier.

**Results.** Figure 2 reports Pass@$N$ results on GSM8K and MATH. On the in-domain dataset GSM8K, we observe that across all backbones and under different numbers of sampled generations, models trained with CFT consistently outperform their SFT counterparts. This demonstrates that focusing updates on critical tokens not only improves single-shot accuracy but also enhances the model's ability to exploit multiple rollouts effectively.

On the more challenging MATH benchmark, our method likewise achieves consistent improvements over SFT across all models. Although the gains are smaller than on GSM8K, this is expected since MATH is more difficult and differs from the training set, making additional diversity harder to translate into correct solutions. These results highlight that CFT remains effective beyond the training distribution.

### 4.4 RL INITIALIZATION WITH CRITICAL-TOKEN FINE-TUNING

We investigate how CFT benefits reinforcement learning (RL) when used as an initialization checkpoint. Specifically, we compare models initialized from SFT and CFT fine-tuning under identical RL configurations using GRPO. The GSM8K and Math training set is used for RL optimization with

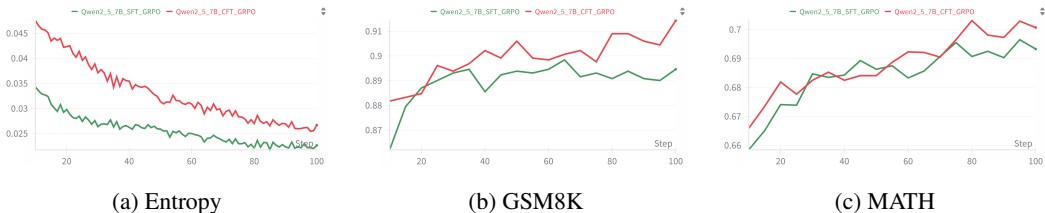

| (a) Entropy | (b) GSM8K | (c) MATH |

Figure 3: Reinforcement learning analysis. CFT-initialized models (Red Line) maintain higher entropy and achieve superior RL performance.

Table 2: Ablation results on Qwen2.5-7B under different definitions of critical tokens. **Avg.** denotes the average accuracy across all 11 benchmarks.

| Methods | Avg. | GSM8K | Math | SVAMP | ASDiv | MAWPS | CARP | TabMWP | Minerva | Gaokao | Olympiad | College |
|---------|------|-------|------|-------|-------|-------|------|--------|---------|--------|----------|---------|
| SFT | 52.8 | 77.0 | 47.3 | 88.0 | 82.4 | 89.7 | 46.2 | 46.3 | 22.4 | 44.2 | 21.6 | 15.4 |
| Union | 57.2 | 81.8 | 55.0 | 88.2 | 84.7 | 92.9 | 50.5 | 52.5 | 24.3 | 45.2 | 28.7 | 25.1 |
| Graded | 56.3 | 82.0 | 53.2 | 90.4 | 85.2 | 92.6 | 49.8 | 48.1 | 23.5 | 46.8 | 27.3 | 20.8 |
| Strict-2 | 58.7 | 83.6 | 58.5 | 90.4 | 85.7 | 93.3 | 51.4 | 53.5 | 25.4 | 47.0 | 30.8 | 26.0 |
| Strict-3 | **59.2** | 81.5 | 59.8 | 90.2 | 85.8 | 92.6 | 53.9 | 53.1 | 26.8 | 48.6 | 28.4 | 30.1 |

the `verl` toolkit,[3] and full RL hyperparameters are listed in Appendix B.2. We track (a) entropy trajectories, (b) GSM8K performance, and (c) MATH performance across training steps (Figure 3).

**CFT-initialized models begin with higher entropy, indicating stronger exploration capacity.** Compared to SFT, CFT checkpoints exhibit substantially larger entropy in their output distributions at the start of RL training. Higher entropy implies that the model maintains multiple plausible reasoning paths instead of collapsing onto a narrow set of solutions. This diversity provides a stronger foundation for RL, allowing models to balance correctness with exploration during optimization.

**SFT models briefly catch up but quickly saturate.** On GSM8K, SFT-initialized models can temporarily reach comparable accuracy to CFT around step 30. However, their entropy rapidly declines, reflecting limited exploration ability. As a result, these models converge prematurely and plateau in performance, leaving little room for further improvement.

**CFT sustains exploration and achieves superior final performance.** In contrast, CFT-initialized models preserve higher entropy throughout training, which enables them to continue exploring diverse reasoning paths. This sustained exploration translates into steady gains beyond 90–100 steps, resulting in higher final accuracy on both GSM8K and MATH.

Overall, these findings highlight that CFT improves RL outcomes not merely by starting from stronger checkpoints, but by equipping models with greater exploration ability, enabling more robust and sustained performance improvements under identical training conditions.

### 4.5 Ablation Study

To better understand the design choices of CFT, we conduct ablations on **Qwen2.5-7B** by varying how critical tokens are defined:

**SFT** trains on all tokens without masking. **Union** ($k = 3$) marks a token critical if replacing it with either of the top-2 or top-3 alternatives breaks correctness. **Graded** ($k = 3$) extends Union with weighted labels: one failed replacement yields weight 1 (weak critical), while two failures yield weight 2 (strong critical). **Strict-2** ($k = 2$) considers only the top-2 alternative; a token is critical if substitution causes failure. **Strict-3** ($k = 3$, **ours**) marks a token as critical only if both top-2 and top-3 substitutions fail. This is the default setting in our main experiments.

**Results.** Table 2 reports the ablation results on Qwen2.5-7B. We find that definitions with stricter criteria consistently outperform looser ones. Union and Graded capture more tokens but dilute the supervision signal, resulting in weaker gains. In contrast, Strict-2 and Strict-3 provide stronger improvements by focusing updates on more decisive tokens. Between them, Strict-3 achieves the

---

[3]https://github.com/volcengine/verl

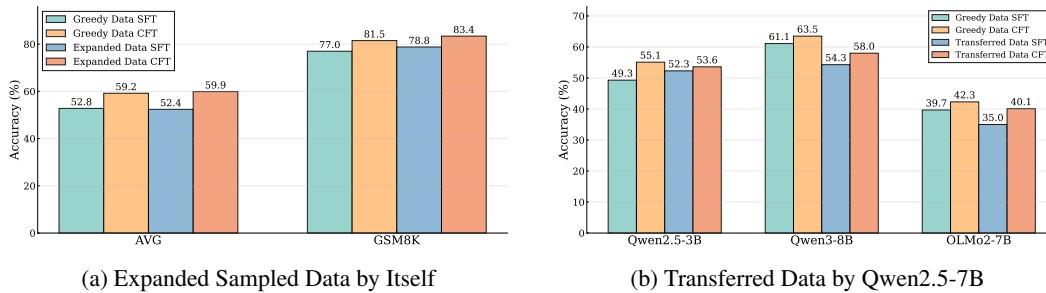

(a) Expanded Sampled Data by Itself        (b) Transferred Data by Qwen2.5-7B

Figure 4: (a) Performance of SFT and CFT when incorporating sampled correct responses. (b) Performance of CFT with critical tokens identified from offline responses. Greedy Data SFT and Greedy Data CFT denote training on each model's own greedy responses (from Section 4.2).

highest overall accuracy (59.2%), while Strict-2 is slightly weaker but remains highly competitive (58.7%), offering a favorable efficiency. Overall, these results validate that precise identification of critical tokens is crucial for effective fine-tuning.

## 5 ANALYSIS

In the previous section, we primarily focused on applying CFT to correct responses generated by the target model itself. While effective, this setting is limited: it cannot directly leverage offline solutions produced by other models, nor can it annotate critical tokens for questions that the target model fails to solve. To address these challenges, this section investigates three aspects: (i) extending CFT to cases where the model cannot solve a question under greedy decoding (Section 5.1); (ii) identifying critical tokens for offline responses not generated by the target model (Section 5.2); and (iii) evaluating whether our method can be applied to domains beyond mathematical reasoning (Section 5.3). Further analyses of the properties of critical tokens are provided in Appendix C.

### 5.1 EXPLOITING SAMPLED RESPONSES

Many training instances cannot be solved correctly under greedy decoding, leaving no usable trajectory for identifying critical tokens. To enlarge the supervision set, we let the model sample candidate solutions at temperature 1 until a correct answer is found (up to 100 attempts). For each correct sampled trajectory $Y = (y_1, \ldots, y_T)$, we assign importance to tokens using the probability gap between a CFT-trained model $M_{\text{cft}}$ (trained in Section 4.2) and its corresponding base model $M_{\text{base}}$:

$$s_t = P_{M_{\text{cft}}}(y_t \mid Q, y_{<t}) - P_{M_{\text{base}}}(y_t \mid Q, y_{<t}).$$

Since $M_{\text{cft}}$ is fine-tuned only on critical tokens, it becomes more sensitive to these positions, assigning them higher probabilities than the base model. We mark the top $15\%$ of tokens with the highest $s_t$ values as critical. This analysis is conducted on Qwen2.5-7B, where 1,334 out of 1,459 GSM8K questions unsolved by greedy decoding are successfully answered through sampling.

In Figure 4a, we compare training only on greedy-correct data against augmenting it with the additional sampled set. The results show that SFT with the enlarged dataset improves in-domain GSM8K accuracy by $+1.8$ points, but slightly lowers average performance across 11 benchmarks ($-0.4$), indicating limited OOD benefit from simply scaling data. In contrast, CFT achieves $59.9\%$ average accuracy, substantially higher than SFT under the same dataset. This suggests that selectively updating on critical tokens enables sampled trajectories to provide effective reasoning supervision, whereas uniform training on all tokens does not.

### 5.2 IDENTIFYING CRITICAL TOKENS FROM OFFLINE RESPONSES

We next investigate whether responses generated by one model can be used to supervise another. Specifically, we take GSM8K solutions from Qwen2.5-7B as external supervision and evaluate three targets: (1) a smaller same-family model, Qwen2.5-3B; (2) a different-version same-family model, Qwen3-8B; and (3) a different-family model, OLMo2-7B.

We consider two transfer settings: **Transferred Data SFT**, where the target model is fine-tuned directly on Qwen2.5-7B responses; and **Transferred Data CFT**, where these responses are first annotated with critical tokens and only those tokens are used for training. For Qwen2.5-3B and Qwen3-8B, which share the same tokenizer as Qwen2.5-7B, we can directly reuse its annotations. For OLMo2-7B, which uses a different tokenizer, the token boundaries do not match those of Qwen2.5-7B, so its critical-token annotations cannot be reused. We adopt a scoring-based strategy (as introduced in Section 5.1) to identify the top $15\%$ most salient tokens.

**Transferred SFT is effective only within the same family.** As shown in Figure 4b, for Qwen2.5-3B, Transferred Data SFT outperforms Greedy Data SFT by $+3.0$ points, showing that responses from a stronger sibling model can provide useful guidance to a weaker one. In contrast, for OLMo2-7B, Transferred Data SFT lags behind Greedy Data SFT by $-4.7$ points, indicating that direct distillation across families is hindered by distribution mismatch.

**Transferred CFT within the same family.** For Qwen2.5-3B and Qwen3-8B, Transferred Data CFT consistently surpasses Transferred Data SFT, and in both cases, it outperforms Greedy Data SFT. This demonstrates that critical tokens identified by Qwen2.5-7B remain informative when transferred to related models in the same family, even when model size or version differs. However, these transferred annotations are still less effective than self-labeled ones (Greedy Data CFT), suggesting that annotations generated by the target model itself provide the strongest supervision.

**Transferred CFT across families.** For OLMo2-7B, applying SFT directly to Qwen-generated responses degrades performance. In contrast, Transferred Data CFT improves average accuracy by $+4.8$ points over Transferred Data SFT, showing that our method can successfully identify salient tokens from external responses and convert them into effective supervision, even when the base responses come from a different model family.

## 5.3 Generalization to the Medical Domain

To assess the applicability of CFT beyond mathematics, we evaluate it on the MedQA (Jin et al., 2021), a multiple-choice dataset constructed from board certification exams. Following the main experimental setup in Section 4.2, we build MedQA training subsets and annotate critical tokens (using $k = 2$ for efficiency). Fine-tuning is conducted on Qwen2.5-7B with both full fine-tuning (Full FT) and LoRA, using the best hyperparameter configurations identified in Section 4.2.

As shown in Table 3, CFT achieves consistent improvements, with gains of $+3.4$ under Full FT and $+0.9$ under LoRA. These results indicate that CFT is not restricted to mathematical reasoning but can also be effectively applied to medical QA. Moreover, our counterfactual perturbation strategy remains effective beyond tasks with verifiable numeric answers,

Table 3: Performance on MedQA.

| Method | Full FT (%) | LoRA (%) |
|--------|-------------|----------|
| SFT    | 46.4        | 46.7     |
| CFT    | **49.8**    | **47.6** |

extending naturally to multiple-choice settings where domain knowledge is essential. Together, these findings suggest that CFT provides a simple way to focus updates on critical tokens across domains, from mathematical reasoning to professional knowledge tasks.

## 6 Conclusion

We introduced **Critical Token Fine-Tuning (CFT)**, a selective-update approach that identifies and updates only tokens crucial for reasoning correctness via counterfactual perturbations. Experiments across multiple LLM families and eleven mathematical reasoning benchmarks demonstrate that CFT achieves higher accuracy than SFT. Importantly, CFT enables improving inference by producing a broader set of high-quality solutions, and serves as an effective initialization for reinforcement learning, sustaining performance gains and maintaining higher entropy during optimization. Beyond mathematics, CFT also proves applicable to other domains, such as medical question answering, highlighting its potential as a general framework for token-level supervision. These results suggest that critical token training offers a widely applicable strategy to enhance LLM reasoning ability, paving the way for future work on more complex reasoning tasks.

## ETHICS STATEMENT

This work focuses on improving fine-tuning methods for large language models, specifically through Critical Token Fine-Tuning (CFT) to enhance reasoning capabilities. We use publicly available datasets, such as GSM8K, which do not involve human subjects or sensitive data. We do not foresee any direct negative impacts from this approach.

## REPRODUCIBILITY STATEMENT

In Section 4.1, we provide a detailed description of the training hyperparameters, datasets, and GPUs used in our experiments. Additionally, our code is available at `https://anonymous.4open.science/r/critical_token-B3D8`.

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

## LIMITATION

Although CFT has shown significant promise in structured tasks such as mathematical reasoning and medical QA, its applicability to unstructured tasks remains an area for future exploration. In particular, tasks involving creativity, such as creative writing, storytelling, pose unique challenges due to the inherent ambiguity and flexibility of their solutions. These tasks often require a creativity that may not always align with the traditional, step-by-step reasoning employed in structured tasks. Therefore, future work could explore how critical token identification can be extended to unstructured tasks.

## A   LLM USAGE

In the preparation of this paper, we only used large language models (LLMs) as an assistive tool for grammar correction and text polishing.

## B   MORE DETAILS ABOUT EXPERIMENT SETUP

### B.1   GREEDY CORRECT SOLUTIONS AND EFFICIENCY OF CRITICAL-TOKEN ANNOTATION

We begin by summarizing the greedy decoding performance of all models on GSM8K (7,473 examples). Table 4 reports, for each model: (i) the number of problems solved correctly under greedy decoding and the corresponding accuracy; (ii) the average length of generated solutions; (iii) the proportion of tokens identified as critical within these correct trajectories; and (iv) the runtime required to annotate 1,000 samples under sequential vs. batched counterfactual rollouts.

As shown in Table 4, although solution lengths differ (e.g., about 240 tokens for Qwen models vs. about 85 for OLMo and Llama), the proportion of critical tokens remains consistently low (6.7%–11.6%), confirming that only a small fraction of positions are indispensable for correctness. Our batched rollout strategy yields dramatic efficiency gains over sequential evaluation. For example, on Qwen2.5-7B, processing 1,000 samples drops from 13.29 hours to 0.50 hours—more than a $25\times$ speedup on 8 A100-80G GPUs. The improvement is particularly pronounced for models with longer outputs (e.g., Qwen), since batched inference amortizes the cost of generating long continuations across positions.

### B.2   RL SETUP

We adopt the GRPO algorithm from the `verl` library to train Qwen models on GSM8K and MATH. During training, we set both the maximum prompt length and response length to 1024 tokens. The overall batch size is 1024 with a per-GPU micro-batch size of 16. The actor is optimized with a learning rate of $1 \times 10^{-6}$, using low-variance KL regularization with coefficient 0.001, and gradient checkpointing is enabled. For each prompt, we sample $n = 5$ rollouts for 15 epochs.

## C   CHARACTERIZING CRITICAL TOKENS

To better understand what types of information CFT emphasizes during fine-tuning, we conduct a detailed analysis of the identified critical tokens. Our goal is to examine their statistical properties, distribution across linguistic categories, and the relative importance of numeric versus non-numeric

Table 4: Greedy decoding statistics and efficiency of critical-token annotation. Parallel rollouts significantly accelerate counterfactual evaluation.

| Model | Greedy Correct | Accuracy (%) | Avg Tokens | Critical Ratio (%) | Sequential /1k (h) | Batched /1k (h) |
|---|---|---|---|---|---|---|
| Qwen2.5-3B | 5,314 | 71.1 | 234.5 | 9.46 | 7.92 | 0.29 |
| Qwen2.5-7B | 6,014 | 80.5 | 221.7 | 7.67 | 13.29 | 0.50 |
| Qwen3-8B | 6,332 | 84.7 | 240.5 | 6.71 | 16.98 | 0.65 |
| Llama3.1-8B | 3,851 | 51.5 | 85.1 | 9.75 | 3.54 | 0.83 |
| OLMo2-7B | 5,283 | 70.7 | 90.0 | 11.61 | 2.01 | 0.29 |

elements. This section provides both quantitative and qualitative perspectives, followed by a controlled experiment on number-only fine-tuning.

## C.1 QUANTITATIVE ANALYSIS

Table 5: Confidence, perplexity, and entropy for critical vs. normal tokens on Qwen2.5-7B, OLMo2-7B, and Llama3.1-8B.

| Metric | Qwen2.5-7B | | OLMo2-7B | | Llama3.1-8B | |
|---|---|---|---|---|---|---|
| | Critical | Normal | Critical | Normal | Critical | Normal |
| Confidence | 0.905 | 0.904 | 0.697 | 0.631 | 0.525 | 0.532 |
| Perplexity | 1.432 | 1.460 | 3.893 | 5.005 | 7.135 | 6.766 |
| Entropy | 0.196 | 0.211 | 0.973 | 1.269 | 1.705 | 1.671 |

## C.2 QUALITATIVE ANALYSIS

Table 5 shows that, even under greedy decoding, token confidence varies substantially across models. Qwen2.5-7B assigns nearly $90\%$ probability to its most likely outputs, whereas Llama3.1-8B rarely exceeds $50\%$, indicating that Qwen is more decisive while LLaMA tends to produce more diverse continuations.

For both Qwen2.5-7B and OLMo2-7B, critical tokens have slightly lower entropy than normal tokens, suggesting a more deterministic choice for these high-impact items. Llama3.1-8B shows a marginal increase in entropy for critical tokens, but the difference is negligible. Overall, confidence, perplexity, and entropy reveal no systematic gap between critical and normal tokens, supporting that our method focuses on a compact yet representative subset.

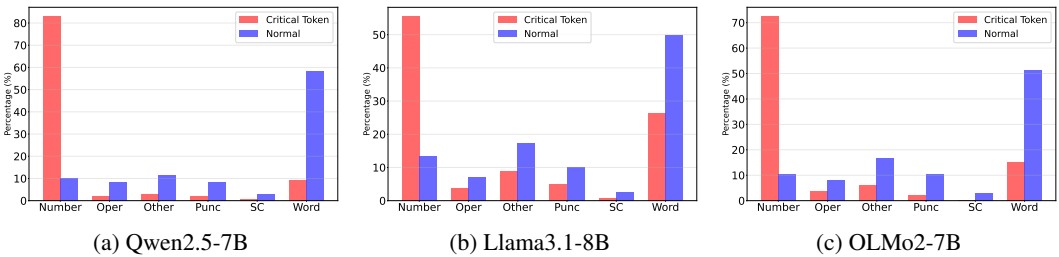

(a) Qwen2.5-7B       (b) Llama3.1-8B       (c) OLMo2-7B

Figure 5: Distributions of token categories for **critical** vs. **normal** tokens across three model families. Categories include numbers, operators (e.g., "+", "-"); punctuation (e.g., ".", ";", "?"); special characters (e.g., "$", "#", "@"); words; and others. Each bar shows the share of a category within either the critical or normal set.

**Qualitative Analysis.** We classify tokens into six categories: numbers, operators, punctuation, special characters, words, and others. Figure 5 compares, for each model, the composition of *critical* tokens and *normal* tokens.

Three observations stand out. First, numbers dominate the pool of critical tokens across models: they account for over 80% of Qwen2.5-7B's critical tokens, over 70% in OLMo2-7B, and over 50%

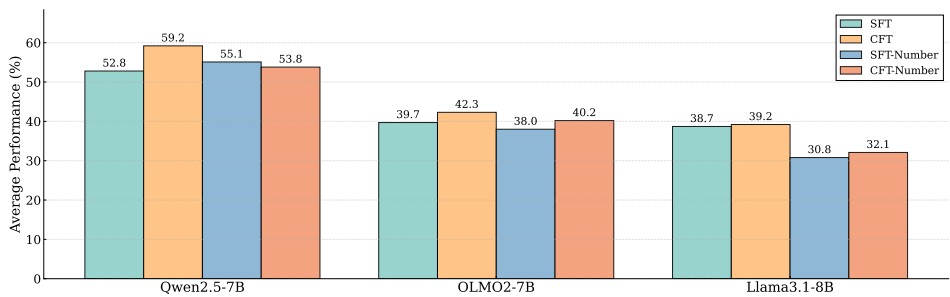

Figure 6: Effect of number-only fine-tuning. This figure illustrates the extent to which numeric tokens alone capture the learning signal across models.

in Llama3.1-8B. Second, words are consistently the second-largest group among critical tokens, contributing roughly 9% for Qwen2.5-7B, 15% for OLMo2-7B, and 25% for Llama3.1-8B. Third, for *normal* tokens, words are by far the most frequent category, exceeding 45% for all models.

These results reveal that while numerical reasoning is central to critical-token selection, lexical cues also play an essential supporting role—especially in OLMo2-7B and Llama3.1-8B, where words occupy a larger share of the critical set.

### C.3 NUMBER-ONLY FINE-TUNING

Given that numbers constitute the majority of critical tokens (Section 5), we investigate whether focusing training exclusively on numeric elements can match the benefits of full critical-token supervision. We compare four settings: (i) fine-tuning on all tokens (SFT), (ii) fine-tuning on the entire set of critical tokens (CFT), (iii) fine-tuning only on numeric tokens drawn from the whole output space (SFT-Number), and (iv) fine-tuning only on numeric tokens within the critical set (CFT-Number). The results, summarized in Figure 6, reveal several trends.

For Qwen2.5-7B, SFT-Number achieves accuracy between SFT and CFT, suggesting that numeric supervision alone captures a substantial portion of the learning signal. CFT-Number remains competitive but does not surpass training on the full critical set, underscoring the added value of non-numeric critical tokens. On OLMo2-7B, the advantage of number-only training becomes modest, while for Llama3.1-8B the gap widens sharply—restricting supervision to numbers incurs large drops relative to both SFT and CFT.

These findings indicate that, although numerical reasoning is a central component of critical-token learning, words et al. items provide indispensable cues, especially for models whose critical sets contain a larger share of words (e.g., 25% in Llama3.1-8B). Limiting fine-tuning to numeric tokens risks discarding these complementary signals, thereby constraining generalization and reasoning depth.

## D ADDITIONALLY RESULTS

Due to space constraints in the main text, we present in Table 6 the complete set of results, including both full fine-tuning and LoRA across all five models and eleven benchmarks.

We observe that the findings reported in Section 4.2 hold consistently under LoRA: (i) updating only critical tokens yields consistent gains over standard SFT; (ii) improvements generalize across in-domain and out-of-domain benchmarks; and (iii) The effect is robust across model families.

This confirms that CFT is effective not only for full fine-tuning but also in parameter-efficient adaptation, further demonstrating its practicality.

Table 6: Comparison of fine-tuning methods on eleven mathematical reasoning benchmarks across five LLMs.

| | Methods | AVG | GSM8K | Math | SVAMP | ASDiv | MAWPS | CARP | TabMWP | Minerva | Gaokao | Olympiad | College |
|---|---|---|---|---|---|---|---|---|---|---|---|---|---|
| **Qwen2.5-3B** | | 43.3 | 65.5 | 34.6 | 80.6 | 74.9 | 81.8 | 38.5 | 37.7 | 12.1 | 29.9 | 10.4 | 10.7 |
| Full | SFT | 49.3 | 69.8 | 43.2 | 81.2 | 78.5 | 85.2 | 44.0 | 46.2 | 15.8 | 39.0 | 18.1 | 20.8 |
| | DFT | 51.1 (+1.8) | 71.9 | 45.0 | 84.5 | 80.8 | 87.1 | 45.2 | 50.7 | 18.4 | 47.4 | 17.9 | 23.5 |
| | Entropy | 50.1 (+0.8) | 68.9 | 45.3 | 82.9 | 79.0 | 85.1 | 45.1 | 49.2 | 16.5 | 38.4 | 18.7 | 22.5 |
| | Attn | 51.5 (+2.2) | 69.4 | 47.5 | 82.3 | 78.1 | 86.1 | 50.0 | 48.4 | 18.4 | 42.1 | 20.7 | 23.6 |
| | CFT | **55.1** (+5.8) | 71.9 | 52.5 | 81.2 | 83.1 | 90.9 | 53.6 | 54.6 | 20.2 | 45.2 | 19.9 | 33.3 |
| LoRA | SFT | 47.8 | 69.5 | 41.9 | 80.5 | 78.4 | 85.5 | 42.4 | 44.4 | 15.1 | 33.0 | 17.3 | 17.7 |
| | DFT | 47.6 (-0.2) | 69.1 | 42.3 | 80.9 | 77.7 | 84.3 | 44.0 | 39.9 | 14.7 | 35.8 | 15.6 | 19.4 |
| | Entropy | 48.5 (+0.7) | 67.2 | 41.7 | 81.3 | 78.9 | 86.8 | 42.9 | 45.3 | 16.5 | 36.1 | 16.7 | 20.5 |
| | Attn | 49.2 (+1.4) | 67.9 | 44.8 | 79.4 | 77.6 | 85.1 | 47.2 | 46.4 | 14.7 | 38.7 | 19.6 | 20.3 |
| | CFT | **53.6** (+5.8) | 73.2 | 49.0 | 84.1 | 83.3 | 91.0 | 47.5 | 52.8 | 17.6 | 42.1 | 20.9 | 27.9 |
| **Qwen2.5-7B** | | 48.3 | 75.2 | 39.6 | 85.5 | 78.7 | 86.4 | 41.4 | 46.0 | 15.4 | 35.8 | 14.8 | 12.0 |
| Full | SFT | 52.8 | 77.0 | 47.3 | 88.0 | 82.4 | 89.7 | 46.2 | 46.3 | 22.4 | 44.2 | 21.6 | 15.4 |
| | DFT | 53.5 (+0.7) | 78.5 | 49.6 | 87.8 | 81.9 | 88.6 | 47.0 | 51.1 | 20.2 | 42.3 | 23.0 | 18.3 |
| | Entropy | 55.4 (+2.6) | 78.6 | 53.9 | 87.9 | 83.4 | 91.7 | 49.2 | 48.6 | 25.4 | 43.9 | 25.3 | 21.6 |
| | Attn | 55.4 (+2.6) | 78.8 | 53.9 | 87.9 | 83.7 | 91.6 | 49.2 | 48.4 | 25.4 | 43.9 | 25.5 | 21.5 |
| | CFT | **59.2** (+6.4) | 81.5 | 59.8 | 90.2 | 85.8 | 92.6 | 53.9 | 53.1 | 26.8 | 48.6 | 28.4 | 30.1 |
| LoRA | SFT | 51.6 | 77.0 | 45.1 | 87.7 | 81.4 | 89.4 | 45.8 | 48.6 | 19.1 | 40.3 | 19.4 | 13.4 |
| | DFT | 52.3 (+0.7) | 77.9 | 45.8 | 89.1 | 82.4 | 89.7 | 45.7 | 48.8 | 19.9 | 41.3 | 19.1 | 15.2 |
| | Entropy | 53.9 (+2.3) | 79.7 | 50.8 | 87.8 | 82.8 | 91.0 | 49.0 | 49.7 | 21.3 | 39.0 | 24.7 | 16.6 |
| | Attn | 53.7 (+2.4) | 79.1 | 50.8 | 87.8 | 82.9 | 91.2 | 49.0 | 49.8 | 21.0 | 37.9 | 24.1 | 17.0 |
| | CFT | **57.6** (+6.0) | 80.3 | 57.8 | 87.7 | 84.1 | 91.3 | 52.0 | 53.2 | 25.7 | 47.3 | 27.6 | 27.1 |
| **Qwen3-8B** | | 60.0 | 84.2 | 66.7 | 90.9 | 83.4 | 89.4 | 54.3 | 46.8 | 28.3 | 52.5 | 36.4 | 26.8 |
| Full | SFT | 61.1 | 83.8 | 69.2 | 90.6 | 84.9 | 91.0 | 54.4 | 45.3 | 31.6 | 56.1 | 36.6 | 28.1 |
| | DFT | 61.0 (-0.1) | 84.5 | 67.8 | 90.7 | 84.2 | 91.0 | 54.6 | 45.5 | 32.0 | 56.6 | 36.7 | 27.5 |
| | Entropy | 60.7 (-0.4) | 84.3 | 67.4 | 91.3 | 84.9 | 91.8 | 54.0 | 46.3 | 29.8 | 54.3 | 36.6 | 26.5 |
| | Attn | 61.1 (+0.0) | 85.7 | 68.9 | 91.2 | 85.0 | 91.5 | 54.4 | 44.3 | 29.8 | 56.4 | 36.3 | 28.1 |
| | CFT | **63.5** (+2.4) | 88.0 | 70.7 | 93.5 | 87.9 | 93.8 | 56.1 | 52.6 | 32.7 | 58.2 | 37.2 | 28.0 |
| LoRA | SFT | 61.3 | 83.9 | 69.6 | 91.3 | 84.7 | 90.1 | 55.9 | 45.5 | 32.0 | 56.1 | 36.3 | 28.8 |
| | DFT | 60.9 (-0.4) | 84.2 | 68.9 | 90.6 | 84.2 | 90.7 | 55.1 | 46.0 | 29.8 | 55.1 | 37.3 | 27.6 |
| | Entropy | 60.9 (-0.4) | 85.0 | 68.4 | 90.4 | 84.3 | 91.5 | 55.1 | 44.8 | 29.4 | 56.6 | 37.0 | 27.8 |
| | Attn | 61.3 (+0.0) | 84.4 | 70.1 | 90.9 | 84.4 | 90.6 | 54.7 | 45.1 | 31.2 | 57.1 | 36.0 | 29.8 |
| | CFT | **63.8** (+2.5) | 86.0 | 73.8 | 93.1 | 86.7 | 92.2 | 57.9 | 51.0 | 32.0 | 59.0 | 38.5 | 31.8 |
| **LLaMA3.1-8B** | | 27.8 | 23.4 | 12.9 | 60.7 | 56.2 | 65.1 | 13.9 | 42.4 | 9.2 | 13.2 | 3.0 | 6.0 |
| Full | SFT | 38.7 | 57.2 | 19.9 | 70.4 | 74.4 | 90.0 | 24.6 | 49.2 | 7.0 | 18.7 | 4.4 | 9.4 |
| | DFT | 38.6 (-0.1) | 57.8 | 19.0 | 71.8 | 74.2 | 91.3 | 21.3 | 48.7 | 8.5 | 18.7 | 4.3 | 8.7 |
| | Entropy | 37.8 (-0.9) | 52.1 | 18.4 | 70.6 | 71.5 | 89.4 | 21.6 | 51.1 | 9.2 | 17.7 | 5.0 | 9.7 |
| | Attn | 36.7 (-2.0) | 60.0 | 16.0 | 66.2 | 72.1 | 87.6 | 20.9 | 48.6 | 7.0 | 15.1 | 4.0 | 6.6 |
| | CFT | **39.2** (+0.5) | 55.6 | 18.9 | 70.1 | 72.7 | 87.4 | 21.9 | 57.2 | 8.5 | 19.7 | 4.7 | 14.1 |
| LoRA | SFT | 39.4 | 53.1 | 19.5 | 70.2 | 74.1 | 89.4 | 24.5 | 60.3 | 7.4 | 19.7 | 5.3 | 9.7 |
| | DFT | 38.8 (-0.6) | 55.2 | 20.0 | 73.1 | 73.9 | 89.6 | 24.3 | 48.9 | 8.8 | 19.5 | 4.3 | 9.4 |
| | Entropy | 38.3 (-1.1) | 52.8 | 18.8 | 69.2 | 73.1 | 88.5 | 21.8 | 55.6 | 8.8 | 19.0 | 4.0 | 10.1 |
| | Attn | 37.0 (-2.4) | 58.5 | 15.7 | 68.6 | 74.3 | 88.5 | 18.8 | 48.4 | 8.1 | 14.8 | 4.3 | 6.8 |
| | CFT | **40.9** (+1.5) | 56.7 | 19.4 | 71.1 | 75.7 | 87.8 | 23.5 | 67.9 | 11.0 | 21.3 | 4.6 | 10.7 |
| **OLMo2-7B** | | 33.2 | 58.2 | 13.3 | 71.3 | 64.6 | 82.3 | 12.5 | 31.3 | 5.5 | 16.1 | 3.9 | 6.1 |
| Full | SFT | 39.7 | 69.1 | 20.9 | 77.0 | 72.8 | 84.4 | 24.1 | 48.2 | 4.8 | 17.9 | 5.2 | 12.6 |
| | DFT | 41.8 (+2.1) | 71.3 | 22.2 | 75.5 | 75.9 | 86.2 | 23.2 | 61.4 | 5.9 | 20.3 | 4.1 | 14.2 |
| | Entropy | 39.3 (-0.4) | 68.3 | 19.3 | 75.3 | 71.7 | 88.9 | 19.1 | 53.4 | 3.7 | 19.7 | 3.1 | 9.3 |
| | Attn | 38.6 (-1.1) | 70.0 | 16.4 | 77.5 | 76.9 | 94.4 | 19.6 | 42.7 | 4.8 | 12.5 | 3.0 | 6.2 |
| | CFT | **42.3** (+2.6) | 74.3 | 21.0 | 75.2 | 78.7 | 92.0 | 20.3 | 61.2 | 4.8 | 21.6 | 4.3 | 12.1 |
| LoRA | SFT | 40.3 | 68.8 | 20.6 | 77.1 | 73.8 | 84.1 | 23.1 | 54.4 | 4.0 | 21.0 | 4.3 | 12.5 |
| | DFT | 41.9 (+1.6) | 70.4 | 21.4 | 75.9 | 76.9 | 89.0 | 22.8 | 61.3 | 5.5 | 20.0 | 4.4 | 13.2 |
| | Entropy | 39.5 (-0.8) | 67.8 | 19.8 | 76.2 | 73.9 | 87.7 | 18.9 | 53.0 | 3.3 | 19.5 | 4.0 | 10.2 |
| | Attn | 38.7 (-1.6) | 71.8 | 17.6 | 76.3 | 76.4 | 94.3 | 18.5 | 42.3 | 4.0 | 15.6 | 2.2 | 6.6 |
| | CFT | **42.8** (+2.5) | 74.8 | 20.8 | 77.1 | 79.9 | 93.3 | 20.5 | 60.0 | 4.4 | 23.6 | 4.0 | 12.2 |