# OpenReview forum: "Enhancing Large Language Model Reasoning via Selective Critical Token Fine-Tuning"
_ICLR.cc/2026/Conference — Submitted to ICLR 2026_

### Official Review · Reviewer_h8JF · 2025-10-31

**Soundness:** 3
**Presentation:** 4
**Contribution:** 2
**Rating:** 4
**Confidence:** 3

**Summary:**

The authors put forward a new SFT method called Critical Token Fine-tuning (CFT). The authors propose only training on critical tokens: a token is deemed non-critical if substituting it with other tokens inside the top-k results in the language model maintaining the correctness of the generated answer using a verifier. The authors test the effectiveness of their SFT method using a suite of small LMs up to 8B parameters using a variety of mathematical reasoning and medical QA tasks. The authors also show its effectiveness for initializing RL training post SFT.

**Strengths:**

* The method is simple and straightforward to implement. Simple methods are often the most effective.
* The presentation of the paper is extremely clear and easy to follow.
* The suite of experiments provided is extensive to demonstrate the benefits of CFT over SFT, albeit no error bars are provided (see weaknesses). The authors consider two different domains: mathematical reasoning and medical QA. The authors consider full finetuning and LoRA adaptation. The authors also consider SFT and RL finetuning. The authors also importantly perform an extensive hyperparameter sweep for all important parameters for SFT.

**Weaknesses:**

* There exists similar methods which measure the importance of specific tokens in a sequence and use this importance to select tokens to contribute to the loss for pre-training, RHO-1 [1]. RHO-1 measures token “importance” or ”criticality” of a token by how different it is to a reference model. So it is not entirely surprising that this can be extended to different notions of “importance”. In this paper’s case it is whether swapping the token for other tokens inside the top-k results in all subsequent predictions being wrong.
* The CFT method is very expensive for all datapoints in the SFT dataset one needs to perform predictions for k counterfactual tokens and at each position t to determine the critical tokens of each sequence. For that compute budget one might be able to run RLFT on top of SFT for the same compute budget as CFT?
* My main issue with the experiments section is the lack of uncertainties in the results. The authors have not run the experiments multiple times to establish error bars over for instance the results in Table 1. For instance the gains in Table 1 for Llama3.1 or OLMo2 could simply be attributed to noise. Likewise in the RL experiments the conclusions are all drawn from experiments over a single seed where the learning curves for GSM and MATH overlap considerably. I find that the evidence does not justify the claims. I understand that repeating an experiment multiple times is expensive in terms of GPU compute required though.

[1] Lin, Zhenghao, et al. "Rho-1: Not all tokens are what you need." arXiv preprint arXiv:2404.07965 (2024).

**Questions:**

* It is not clear to me how parallel critical identification reduces computation time in lines 157-161. Is it due to KV-caching?
* The claim that CFT initialized models have higher entropy and therefore encourage exploration is very interesting. I’m assuming the entropy is in the LLMs logits? Should this not result in worse performance, since the entropy of the logits is what we are trying to minimize when we are minimizing next token predictions? An increase in exploration should result in new states being visited by the LLM. So surely this should result in more diverse predictions: increased pass@k? It is not clear to me what evidence you have for stronger exploration in the paper.

---

> ### Author Response · Authors · 2025-11-18
> **Rebuttal 1**
>
> ## Weaknesses
>
> **1. There exists similar methods which measure the importance of specific tokens in a sequence and use this importance to select tokens to contribute to the loss for pre-training, RHO-1 [1]. RHO-1 measures token “importance” or ”criticality” of a token by how different it is to a reference model. So it is not entirely surprising that this can be extended to different notions of “importance”. In this paper’s case it is whether swapping the token for other tokens inside the top-k results in all subsequent predictions being wrong.**
>
> **Response 1:**
>
> While our method shares the high-level idea of token-level selection with RHO-1, the **design motivation, supervision mechanism, and empirical behavior** differ fundamentally. We summarize the distinctions and our contributions as follows.
>
> - **Different training stage, different supervision signal.** RHO-1 operates in the *pre-training* stage and defines token importance through loss difference to a reference model.  CFT operates in the SFT stage and identifies criticality through counterfactual functional necessity; a token is updated only if *every* top-k substitution causes answer failure.  This shifts the notion of “importance” from *distributional deviation* to *causal indispensability for correctness*, which is a different objective.
>
> - **No auxiliary model, true sparse-token training.** RHO-1, TIS-DPO / cDPO require two auxiliary models; DFT still update all tokens.  CFT needs no extra models, and updates only ~12% of tokens.  This sparsity is not seen in prior SFT-stage methods, yet consistently outperforms full SFT across all backbones.
>
> - **Benefits beyond accuracy: diversity and RL initialization.** Because only indispensable tokens are updated, non-critical tokens retain higher entropy.  CFT maintains higher entropy throughout RL and achieves higher final accuracy, whereas SFT-trained reference models used by RHO-1 tend to collapse early.
>
> - **Direct empirical comparison.** We followed the original RHO-1 setup to SFT stage and used each model’s SFT checkpoint (Table 1 in the paper) as the reference model.  Across all five backbones, CFT consistently outperforms RHO-1:
>
> | Model | Method | Avg | GSM8K | Math | SVAMP | ASDiv | MAWPS | CARP | TabMWP | Minerva | Gaokao | Olympiad | College |
> | - | - | - | - | - | - | - | - | - | - | - | - | - | - |
> | Qwen2.5-3B | Rho-1 | 49.8 | 69.1 | 43.2 | 82.9 | 78.9 | 84.9 | 46.4 | 49.4 | 15.8 | 38.4 | 17.5 | 21.2 |
> | Qwen2.5-3B | Ours | 55.1 | 71.9 | 52.5 | 81.2 | 83.1 | 90.9 | 53.6 | 54.6 | 20.2 | 45.2 | 19.9 | 33.3 |
> | Qwen2.5-7B | Rho-1 | 52.6 | 77.3 | 47.8 | 88.7 | 81.9 | 89.8 | 47.1 | 46.9 | 20.6 | 41.0 | 22.5 | 15.0 |
> | Qwen2.5-7B | Ours | 59.2 | 81.5 | 59.8 | 90.2 | 85.8 | 92.6 | 53.9 | 53.1 | 26.8 | 48.6 | 28.4 | 30.1 |
> | Qwen3-8B | Rho-1 | 60.7 | 83.2 | 68.9 | 91.5 | 83.9 | 91.3 | 53.4 | 45.5 | 31.6 | 55.1 | 35.7 | 27.8 |
> | Qwen3-8B | Ours | 63.5 | 88.0 | 70.7 | 93.5 | 87.9 | 93.8 | 56.1 | 52.6 | 32.7 | 58.2 | 37.2 | 28.0 |
> | Llama3.1-8B | Rho-1 | 37.8 | 56.6 | 18.1 | 65.9 | 72.7 | 90.2 | 22.6 | 47.6 | 9.2 | 18.2 | 4.3 | 10.1 |
> | Llama3.1-8B | Ours | 39.2 | 55.6 | 18.9 | 70.1 | 72.7 | 87.4 | 21.9 | 57.2 | 8.5 | 19.7 | 4.7 | 14.1 |
> | OLMo2-7B | Rho-1 | 40.6 | 69.2 | 19.5 | 76.8 | 75.6 | 91.7 | 21.9 | 46.5 | 7.7 | 21.3 | 4.3 | 12.4 |
> | OLMo2-7B | Ours | 42.3 | 74.3 | 21.0 | 75.2 | 78.7 | 92.0 | 20.3 | 61.2 | 4.8 | 21.6 | 4.3 | 12.1 |
>
>
> Although the idea of measuring token importance has precedents, our work introduces a new formulation at the SFT stage, requiring no auxiliary model, and producing large, consistent improvements in accuracy, diversity, and RL performance.  We will incorporate the clarified comparison and full results table into the revised version.

---

> > ### Author Response · Authors · 2025-11-18
> > **Rebuttal 2**
> >
> > ## Weaknesses
> >
> > **2. The CFT method is very expensive for all datapoints in the SFT dataset one needs to perform predictions for k counterfactual tokens and at each position t to determine the critical tokens of each sequence. For that compute budget one might be able to run RLFT on top of SFT for the same compute budget as CFT?**
> >
> > **Response 2:**
> >
> > We clarify two points:
> > (1) CFT serves a different role than RLFT, and
> > (2) CFT can be made highly efficient through lightweight variants.
> >
> > ---
> >
> > **1.CFT and RLFT address different needs.**  CFT improves the SFT checkpoint by preserving entropy and reducing mode collapse, which provides a stronger initialization for RL.  As shown in Section 4.4 (Figure 3), CFT-initialized models begin RL with higher entropy, maintain exploration longer, and achieve higher final GSM8K/MATH accuracy than RL initialized from SFT. This observation is consistent with Reviewer 33GQ's positive assessment:  **CFT-initialized checkpoints begin with higher entropy for RL training and sustain exploration. With a better initialization, models could be trained with even more steps.**
> > It supports our claim that CFT complements rather than replaces RLFT.
> >
> >
> > **2.CFT can be made lightweight.** While full counterfactual substitution is expensive, we introduce a simple and efficient two-stage variant in this rebuttal:
> >
> > (1) Annotate only ~500 examples using standard CFT and briefly fine-tune the model to obtain $ M_{\text{cft}} $.
> > (2) For the full dataset, compute the probability gap between $ M_{\text{cft}} $ and $ M_{\text{base}} $ to score token importance like **Section 5.1 Exploiting Sampled Responses**:
> >
> > $$
> > s_t = P_{M_{\text{cft}}}(y_t \mid Q, y_{<t}) - P_{M_{\text{base}}}(y_t \mid Q, y_{<t}).
> > $$
> >
> > This eliminates most rollouts while preserving the core benefits of CFT. Across the five models, generating 500 annotated samples takes on average **15.6 minutes** on 8×A100 GPUs. Notably, once the 500-sample CFT model is obtained, it can annotate new data extremely efficiently: generating 1000 additional annotations takes only **1.2 minutes on average** on 1×A100 GPUs. The efficient variant remains highly competitive across all backbones and even surpasses the standard version on Qwen2.5-7B.
> >
> > | Model | Method | AVG | GSM8K | MATH | SVAMP | ASDiv | MAWPS | CARP | TabMWP | Minerva | Gaokao | Olympiad | College |
> > |-------|--------|-----|--------|--------|--------|--------|--------|--------|----------|---------|---------|------------|-----------|
> > | Qwen2.5-3B | SFT | 49.3 | 69.8 | 43.2 | 81.2 | 78.5 | 85.2 | 44.0 | 46.2 | 15.8 | 39.0 | 18.1 | 20.8 |
> > | Qwen2.5-3B | Standard CFT | 55.1 | 71.9 | 52.5 | 81.2 | 83.1 | 90.9 | 53.6 | 54.6 | 20.2 | 45.2 | 19.9 | 33.3 |
> > | Qwen2.5-3B | Efficient CFT | 54.9 | 73.8 | 51.9 | 85.7 | 82.4 | 89.6 | 51.0 | 55.2 | 20.2 | 45.5 | 19.7 | 29.0 |
> > | Qwen2.5-7B | SFT | 52.8 | 77.0 | 47.3 | 88.0 | 82.4 | 89.7 | 46.2 | 46.3 | 22.4 | 44.2 | 21.6 | 15.4 |
> > | Qwen2.5-7B | Standard CFT | 59.2 | 81.5 | 59.8 | 90.2 | 85.8 | 92.6 | 53.9 | 53.1 | 26.8 | 48.6 | 28.4 | 30.1 |
> > | Qwen2.5-7B | Efficient CFT | 63.2 | 84.5 | 67.8 | 91.5 | 87.0 | 93.2 | 56.6 | 56.0 | 35.7 | 57.1 | 31.3 | 34.5 |
> > | Qwen3-8B | SFT | 61.1 | 83.8 | 69.2 | 90.6 | 84.9 | 91.0 | 54.4 | 45.3 | 31.6 | 56.1 | 36.6 | 28.1 |
> > | Qwen3-8B | Standard CFT | 63.5 | 88.0 | 70.7 | 93.5 | 87.9 | 93.8 | 56.1 | 52.6 | 32.7 | 58.2 | 37.2 | 28.0 |
> > | Qwen3-8B | Efficient CFT | 62.4 | 87.8 | 71.1 | 92.4 | 86.3 | 92.7 | 54.2 | 46.6 | 33.8 | 55.6 | 37.9 | 27.9 |
> > | Llama3.1-8B | SFT | 38.7 | 57.2 | 19.9 | 70.4 | 74.4 | 90.0 | 24.6 | 49.2 | 7.0 | 18.7 | 4.4 | 9.4 |
> > | Llama3.1-8B | Standard CFT | 39.2 | 55.6 | 18.9 | 70.1 | 72.7 | 87.4 | 21.9 | 57.2 | 8.5 | 19.7 | 4.7 | 14.1 |
> > | Llama3.1-8B | Efficient CFT | 38.1 | 56.9 | 18.1 | 70.2 | 72.5 | 89.5 | 23.1 | 46.7 | 10.7 | 17.9 | 4.0 | 9.3 |
> > | OLMo2-7B | SFT | 39.7 | 69.1 | 20.9 | 77.0 | 72.8 | 84.4 | 24.1 | 48.2 | 4.8 | 17.9 | 5.2 | 12.6 |
> > | OLMo2-7B | Standard CFT | 42.3 | 74.3 | 21.0 | 75.2 | 78.7 | 92.0 | 20.3 | 61.2 | 4.8 | 21.6 | 4.3 | 12.1 |
> > | OLMo2-7B | Efficient CFT | 42.4 | 72.2 | 21.2 | 75.9 | 78.5 | 92.3 | 23.6 | 58.7 | 4.8 | 21.0 | 4.3 | 13.5 |
> >
> > ---
> >
> >
> >
> > In summary, CFT provides a stronger RL initialization due to its entropy-preserving behavior, and the lightweight variants make the computation practical.

---

> > > ### Author Response · Authors · 2025-11-18
> > > **Rebuttal 3**
> > >
> > > ## Weakness
> > >
> > > **3. My main issue with the experiments section is the lack of uncertainties in the results. The authors have not run the experiments multiple times to establish error bars over for instance the results in Table 1. For instance the gains in Table 1 for Llama3.1 or OLMo2 could simply be attributed to noise. Likewise in the RL experiments the conclusions are all drawn from experiments over a single seed where the learning curves for GSM and MATH overlap considerably. I find that the evidence does not justify the claims. I understand that repeating an experiment multiple times is expensive in terms of GPU compute required though.**
> > >
> > > **Response 3:**
> > >
> > > To address this, we conducted additional experiments across multiple random seeds for both SFT and CFT, and repeated the RL experiments twice to quantify variance. Across all evaluations, CFT consistently outperforms SFT. The improvements are stable across seeds and exceed the observed variance, confirming that the gains are not noise-driven.
> > >
> > > **Supervised fine-tuning reproducibility (three seeds: 0 / 42 / 2025)**
> > >
> > > Llama3.1-8B (CFT vs. SFT)
> > >
> > > | Model | Seed | AVG | GSM8K | MATH | SVAMP | ASDiv | MAWPS | CARP | TabMWP | Minerva | Gaokao | Olympiad | College |
> > > | - | - | - | - | - | - | - | - | - | - | - | - | - | - |
> > > | Llama3.1-8B-CFT | 0 | 39.6 | 57.4 | 19.5 | 69.0 | 72.8 | 85.5 | 23.1 | 60.2 | 7.0 | 17.9 | 4.1 | 13.9 |
> > > |  | 42 | 39.2 | 55.6 | 18.9 | 70.1 | 72.7 | 87.4 | 21.9 | 57.2 | 8.5 | 19.7 | 4.7 | 14.1 |
> > > |  | 2025 | 39.5 | 57.8 | 19.2 | 69.9 | 74.0 | 84.4 | 20.4 | 63.0 | 8.1 | 18.7 | 5.0 | 13.8 |
> > > | Llama3.1-8B-SFT | 0 | 38.9 | 57.0 | 20.0 | 71.6 | 74.5 | 90.4 | 24.3 | 48.9 | 5.5 | 20.8 | 5.2 | 9.4 |
> > > |  | 42 | 38.7 | 57.2 | 19.9 | 70.4 | 74.4 | 90.0 | 24.6 | 49.2 | 7.0 | 18.7 | 4.4 | 9.4 |
> > > |  | 2025 | 39.0 | 57.2 | 20.2 | 71.9 | 74.3 | 90.3 | 25.1 | 49.5 | 6.6 | 19.7 | 4.7 | 9.8 |
> > >
> > >
> > > OLMo2-7B (CFT vs. SFT)
> > >
> > > | Model | Seed | AVG | GSM8K | MATH | SVAMP | ASDiv | MAWPS | CARP | TabMWP | Minerva | Gaokao | Olympiad | College |
> > > | - | - | - | - | - | - | - | - | - | - | - | - | - | - |
> > > | OLMo-CFT | 0 | 42.3 | 73.5 | 20.9 | 76.0 | 79.5 | 92.6 | 21.1 | 59.6 | 5.1 | 20.5 | 4.4 | 11.7 |
> > > |  | 42 | 42.3 | 74.3 | 21.0 | 75.2 | 78.7 | 92.0 | 20.3 | 61.2 | 4.8 | 21.6 | 4.3 | 12.1 |
> > > |  | 2025 | 41.6 | 72.6 | 21.5 | 73.0 | 77.3 | 89.2 | 21.4 | 59.3 | 5.1 | 21.3 | 4.4 | 12.6 |
> > > | OLMo-SFT | 0 | 40.1 | 69.3 | 20.9 | 77.8 | 73.6 | 85.7 | 25.6 | 47.9 | 5.1 | 17.4 | 5.5 | 12.4 |
> > > |  | 42 | 39.7 | 69.1 | 20.9 | 77.0 | 72.8 | 84.4 | 24.1 | 48.2 | 4.8 | 17.9 | 5.2 | 12.6 |
> > > |  | 2025 | 39.7 | 68.8 | 20.8 | 76.9 | 73.1 | 85.3 | 23.3 | 47.4 | 4.4 | 19.7 | 4.9 | 12.4 |
> > >
> > > ---------------------------------------------------------------------
> > >
> > > **Seed-level variability**
> > >
> > > | Model | Method | Mean ± Std |
> > > | - | - | - |
> > > | Llama3.1-8B | CFT | 39.42 ± 0.05 |
> > > | Llama3.1-8B | SFT | 38.85 ± 0.03 |
> > > | OLMo2-7B | CFT | 42.07 ± 0.17 |
> > > | OLMo2-7B | SFT | 39.84 ± 0.05 |
> > >
> > > The differences between CFT and SFT are significantly larger than the seed-level variance.
> > >
> > >
> > > **RL reproducibility (Qwen2.5-7B, repeated twice)**
> > > RL involves stochastic sampling with temperature, so we retrained both SFT-initialized and CFT-initialized RL runs twice. The results below include all attempts and the original paper runs.
> > >
> > > GSM8K
> > >
> > > | Step | SFT-0 | CFT-0 | SFT-1 | CFT-1 | SFT-Paper | CFT-Paper |
> > > | - | - | - | - | - | - | - |
> > > | 20 | 89.01 | 88.70 | 88.93 | 88.56 | 88.70 | 88.48 |
> > > | 40 | 89.39 | 89.99 | 89.01 | 89.39 | 88.55 | 90.22 |
> > > | 60 | 89.23 | 90.45 | 88.93 | 90.75 | 89.46 | 89.84 |
> > > | 80 | 90.30 | 89.92 | 89.69 | 89.92 | 89.08 | 90.90 |
> > > | 100 | 90.07 | 90.37 | 89.00 | 90.60 | 89.46 | 91.43 |
> > >
> > > MATH
> > >
> > > | Step | SFT-0 | CFT-0 | SFT-1 | CFT-1 | SFT-Paper | CFT-Paper |
> > > | - | - | - | - | - | - | - |
> > > | 20 | 68.29 | 67.97 | 67.09 | 68.31 | 67.41 | 68.19 |
> > > | 40 | 68.03 | 68.27 | 67.63 | 68.91 | 68.43 | 68.25 |
> > > | 60 | 68.15 | 68.59 | 68.09 | 69.09 | 68.33 | 69.23 |
> > > | 80 | 69.03 | 69.21 | 68.13 | 70.19 | 69.07 | 70.31 |
> > > | 100 | 68.81 | 69.07 | 68.73 | 70.05 | 69.33 | 70.07 |
> > >
> > >
> > > **Averaged variance**
> > >
> > > | Metric | SFT Avg | CFT Avg | SFT Var | CFT Var |
> > > | - | - | - | - | - |
> > > | GSM8K-Step20 | 88.88 | 88.58 | 0.0259 | 0.0124 |
> > > | GSM8K-Step40 | 88.98 | 89.87 | 0.1769 | 0.1836 |
> > > | GSM8K-Step60 | 89.21 | 90.35 | 0.0706 | 0.2150 |
> > > | GSM8K-Step80 | 89.69 | 90.30 | 0.3721 | 0.2786 |
> > > | GSM8K-Step100 | 89.51 | 90.80 | 0.2881 | 0.3109 |
> > > | MATH-Step20 | 67.60 | 68.16 | 0.3861 | 0.0297 |
> > > | MATH-Step40 | 68.03 | 68.48 | 0.1600 | 0.1409 |
> > > | MATH-Step60 | 68.19 | 68.97 | 0.0156 | 0.1132 |
> > > | MATH-Step80 | 68.74 | 69.90 | 0.2825 | 0.3641 |
> > > | MATH-Step100 | 68.96 | 69.73 | 0.1061 | 0.3268 |
> > >
> > > CFT maintains consistently higher accuracy in most steps, and the magnitude of improvement exceeds the observed variance. Therefore, the observed benefits of CFT are not artifacts of random noise but reflect genuine improvements in both SFT and RL training.

---

> > > > ### Author Response · Authors · 2025-11-18
> > > > **Rebuttal 4**
> > > >
> > > > ## Questions
> > > > **1. It is not clear to me how parallel critical identification reduces computation time in lines 157-161. Is it due to KV-caching?**
> > > >
> > > > **Response 1:**
> > > >
> > > > It is not due to KV caching. The speedup comes from batching all counterfactual rollouts for one example into a single forward pass, which enables much better utilization of vLLM’s throughput. Averaged across all five models, the optimized pipeline achieves more than a 15× speedup over sequential processing.
> > > >
> > > > Sequential corresponds to the standard vLLM workflow: we submit full examples to vLLM (batch size automatically determined by vLLM), and vLLM evaluates counterfactuals by scanning token positions one by one. For each token position, vLLM simultaneously processes that position for all samples in the batch, then moves to the next position.
> > > >
> > > > Parallel critical identification corresponds to our optimized pipeline: we process one example at a time, first run a single forward pass to obtain its top-2/3 alternatives for all token positions, and then batch all counterfactual prefixes of that single example. The batch size is again controlled automatically by vLLM, which evaluates all prefixes in one shot.
> > > >
> > > > **2. The claim that CFT initialized models have higher entropy and therefore encourage exploration is very interesting. I’m assuming the entropy is in the LLMs logits? Should this not result in worse performance, since the entropy of the logits is what we are trying to minimize when we are minimizing next token predictions? An increase in exploration should result in new states being visited by the LLM. So surely this should result in more diverse predictions: increased pass@k? It is not clear to me what evidence you have for stronger exploration in the paper.**
> > > >
> > > > **Response 2:**
> > > >
> > > > We clarify below why higher entropy does not harm performance, how it relates to exploration, and what evidence supports our claim.
> > > >
> > > > - First, entropy here refers to the entropy of the model’s output logits, which is a standard proxy for exploration in both LLMs and RL. Prior work shows that maintaining sufficient entropy is crucial for preventing early collapse and enabling robust search over the solution space [1,2].
> > > >
> > > > - Second, higher entropy does **not** imply lower accuracy during supervised fine-tuning. CFT minimizes loss only on *critical* tokens; non-critical tokens are not forced to collapse toward a single distributional mode. As a result, CFT preserves logit diversity without affecting correctness, allowing the model to both (1) match or exceed SFT accuracy and (2) maintain higher entropy.
> > > >
> > > > - Third, pass@k provides direct behavioral evidence for exploration. A model with high diversity but low quality will generate many different answers, but most of them are incorrect, resulting in low pass@k. Conversely, a model with low diversity but high quality tends to repeat the same answer across samples, which limits the improvement in pass@k even when pass@1 is already strong. This perspective aligns with recent work on pass@k and exploration [3, 4].
> > > >
> > > > Most importantly, **Figure 3(a) in our paper provides explicit evidence**:  CFT-initialized models exhibit **consistently higher entropy throughout the entire RL training process**. This directly demonstrates that CFT maintains a more exploratory policy and avoids premature convergence.
> > > >
> > > >
> > > >
> > > > [1] Preserving Diversity in Supervised Fine-Tuning of Large Language Models, ICLR 2025
> > > >
> > > > [2] State Entropy Regularization for Robust Reinforcement Learning, NeurIPS 2025
> > > >
> > > > [3] Pass@k Training for Adaptively Balancing Exploration and Exploitation of Large Reasoning Models, arXiv:2508.10751
> > > >
> > > > [4] Pass@K Policy Optimization: Solving Harder Reinforcement Learning Problems, NeurIPS 2025

---

> ### Author Response · Authors · 2025-11-26
>
> Dear Reviewer h8JF,
>
> Wishing you a happy and blessed Thanksgiving!
>
> Thank you for your detailed and insightful review. We have added additional experiments and analyses to address all the questions you raised. If you have any further suggestions or would like clarification on any part of our response, please feel free to let us know. We would be very happy to continue the discussion.
>
> We truly appreciate the time and effort you have devoted to reviewing our work.
>
> Best regards,
>
> The Authors

---

### Official Review · Reviewer_4YV3 · 2025-10-31

**Soundness:** 2
**Presentation:** 2
**Contribution:** 2
**Rating:** 4
**Confidence:** 3

**Summary:**

The paper presents the idea of critical token fine-tuning, based on the hypothesis that only a small fraction of tokens actually determines reasoning correctness. Compared with standard supervised fine-tuning approach, the paper proposes critical token fine-tuning, which first finds critical tokens through counterfactual perturbation, followed by fine-tuning only on these critical positions.

**Strengths:**

- The paper proposes a novel method and observation that takes into account token importance for reasoning to perform more targeted supervised fine-tuning
- The idea is simple and elegant, and the empirical results demonstrates that fine-tuning on a small fraction of tokens can outperform standard SFT methods

**Weaknesses:**

- the counterfactual computation in the first stage for token importance calculation can be computationally heavy
- it is unclear if the reasoning correctness is attributable to individual tokens independently? Reasoning may usually depend on a set of inter-dependent tokens and hence the critical token assumption could be an over-simplified assumption.

**Questions:**

- Does the method account for inter-token dependencies, e.g., a pair, or a set of tokens that together determine correctness even if each token alone is replaceable?
- Does perturbation of multiple non-critical tokens affect solution quality?

---

> ### Author Response · Authors · 2025-11-18
> **Rebuttal 1**
>
> ## Weaknesses
>
> **1. the counterfactual computation in the first stage for token importance calculation can be computationally heavy**
>
> **Response 1:**
>
> While naïve counterfactual rollouts can be expensive, our implementation already incorporates an efficient batched evaluation strategy, and we further propose a lightweight alternative:
>
> (1) Annotate only ~500 examples using standard CFT and briefly fine-tune the model to obtain $ M_{\text{cft}} $.
> (2) For the full dataset, compute the probability gap between $ M_{\text{cft}} $ and $ M_{\text{base}} $ to score token importance like **Section 5.1 Exploiting Sampled Responses**:
>
> $$
> s_t = P_{M_{\text{cft}}}(y_t \mid Q, y_{<t}) - P_{M_{\text{base}}}(y_t \mid Q, y_{<t}).
> $$
>
> This avoids full counterfactual rollouts for the entire dataset. Across the five models, generating 500 annotated samples takes on average **15.6 minutes** on 8×A100 GPUs. Notably, once the 500-sample CFT model is obtained, it can annotate new data extremely efficiently: generating 1000 additional annotations takes only **1.2 minutes on average** on 1×A100 GPUs.
>
>
> **Performance of Efficient CFT**
>
> | Model | Method | AVG | GSM8K | MATH | SVAMP | ASDiv | MAWPS | CARP | TabMWP | Minerva | Gaokao | Olympiad | College |
> |-------|--------|-----|--------|--------|--------|--------|--------|--------|----------|---------|---------|------------|-----------|
> | Qwen2.5-3B | SFT | 49.3 | 69.8 | 43.2 | 81.2 | 78.5 | 85.2 | 44.0 | 46.2 | 15.8 | 39.0 | 18.1 | 20.8 |
> | Qwen2.5-3B | Standard CFT | 55.1 | 71.9 | 52.5 | 81.2 | 83.1 | 90.9 | 53.6 | 54.6 | 20.2 | 45.2 | 19.9 | 33.3 |
> | Qwen2.5-3B | Efficient CFT | 54.9 | 73.8 | 51.9 | 85.7 | 82.4 | 89.6 | 51.0 | 55.2 | 20.2 | 45.5 | 19.7 | 29.0 |
> | Qwen2.5-7B | SFT | 52.8 | 77.0 | 47.3 | 88.0 | 82.4 | 89.7 | 46.2 | 46.3 | 22.4 | 44.2 | 21.6 | 15.4 |
> | Qwen2.5-7B | Standard CFT | 59.2 | 81.5 | 59.8 | 90.2 | 85.8 | 92.6 | 53.9 | 53.1 | 26.8 | 48.6 | 28.4 | 30.1 |
> | Qwen2.5-7B | Efficient CFT | 63.2 | 84.5 | 67.8 | 91.5 | 87.0 | 93.2 | 56.6 | 56.0 | 35.7 | 57.1 | 31.3 | 34.5 |
> | Qwen3-8B | SFT | 61.1 | 83.8 | 69.2 | 90.6 | 84.9 | 91.0 | 54.4 | 45.3 | 31.6 | 56.1 | 36.6 | 28.1 |
> | Qwen3-8B | Standard CFT | 63.5 | 88.0 | 70.7 | 93.5 | 87.9 | 93.8 | 56.1 | 52.6 | 32.7 | 58.2 | 37.2 | 28.0 |
> | Qwen3-8B | Efficient CFT | 62.4 | 87.8 | 71.1 | 92.4 | 86.3 | 92.7 | 54.2 | 46.6 | 33.8 | 55.6 | 37.9 | 27.9 |
> | Llama3.1-8B | SFT | 38.7 | 57.2 | 19.9 | 70.4 | 74.4 | 90.0 | 24.6 | 49.2 | 7.0 | 18.7 | 4.4 | 9.4 |
> | Llama3.1-8B | Standard CFT | 39.2 | 55.6 | 18.9 | 70.1 | 72.7 | 87.4 | 21.9 | 57.2 | 8.5 | 19.7 | 4.7 | 14.1 |
> | Llama3.1-8B | Efficient CFT | 38.1 | 56.9 | 18.1 | 70.2 | 72.5 | 89.5 | 23.1 | 46.7 | 10.7 | 17.9 | 4.0 | 9.3 |
> | OLMo2-7B | SFT | 39.7 | 69.1 | 20.9 | 77.0 | 72.8 | 84.4 | 24.1 | 48.2 | 4.8 | 17.9 | 5.2 | 12.6 |
> | OLMo2-7B | Standard CFT | 42.3 | 74.3 | 21.0 | 75.2 | 78.7 | 92.0 | 20.3 | 61.2 | 4.8 | 21.6 | 4.3 | 12.1 |
> | OLMo2-7B | Efficient CFT | 42.4 | 72.2 | 21.2 | 75.9 | 78.5 | 92.3 | 23.6 | 58.7 | 4.8 | 21.0 | 4.3 | 13.5 |
>
> ---
>
> These results confirm that CFT is computationally feasible while maintaining strong performance across all models, and the efficient variant offers a practical alternative with minimal cost.
>
>
>
> **2. it is unclear if the reasoning correctness is attributable to individual tokens independently? Reasoning may usually depend on a set of inter-dependent tokens and hence the critical token assumption could be an over-simplified assumption.**
>
> **Response 2:**
>
> CFT does **not** assume independence. Each token is evaluated under the **full prefix** and its perturbation triggers **full suffix regeneration**, meaning its effect is assessed within the complete reasoning context. To directly examine multi-token dependencies, we performed a pair-level analysis across five model families.
>
> ---
>
> **Pair-level empirical results**
>
> The table below shows the percentage of critical tokens forming contiguous groups.
>
> | Model | Independent (%) | Paired (%) |
> |---|---|---|
> | Qwen2.5-3B | 45.85 | 54.15 |
> | Qwen2.5-7B | 47.59 | 52.41 |
> | Qwen3-8B | 46.82 | 53.18 |
> | Llama3.1-8B | 66.91 | 33.09 |
> | OLMo2-7B | 77.66 | 22.34 |
>
> **Paired critical tokens are common**, especially in the Qwen family, where paired tokens slightly exceed independent ones. These results show that multi-token dependencies **exist but are already captured by CFT**.

---

> > ### Author Response · Authors · 2025-11-18
> > **Rebuttal 2**
> >
> > ## Questions
> > **1. Does the method account for inter-token dependencies, e.g., a pair, or a set of tokens that together determine correctness even if each token alone is replaceable?**
> >
> > **Response 1:**
> >
> > Yes.  CFT implicitly captures such dependencies because:
> >
> > - **Prefix conditioning** ensures each token is evaluated under the complete context.
> > - **Our pair-level analysis (above)** empirically confirms that multi-token dependencies are short and localized, and effectively detected by single-token perturbations.
> >
> > Thus, the method is compatible with inter-token dependency structures.
> >
> >
> >
> > **2. Does perturbation of multiple non-critical tokens affect solution quality?**
> >
> > **Response 2:**
> >
> > In an autoregressive model, every token is generated conditioned on the entire prefix. Thus, the idea of “perturbing multiple tokens simultaneously” does not align with the model’s generative process. For example, consider a reasoning chain of length 10 with tokens $t_1, \ldots, t_{10}$. If we attempt to perturb $t_2, t_3$, and $t_5$, once $t_3$ is perturbed, the model will regenerate the suffix, and the original $t_5$ will no longer exist in that trajectory. Therefore, multi-token parallel perturbations are not well-defined.
> >
> > Under this structure, CFT’s single-token counterfactual already reveals multi-token vulnerabilities: if a reasoning step relies on a set of fragile tokens, perturbing *any* member breaks the downstream chain during suffix regeneration and is detected. Moreover, our pair-level analysis (table above) shows that multi-token dependencies exist but are already captured by CFT.

---

> ### Author Response · Authors · 2025-11-26
>
> Dear Reviewer 4YV3,
>
> Wishing you a happy and blessed Thanksgiving!
>
> Thank you for your detailed and insightful review. We have added additional experiments and analyses to address all the questions you raised. If you have any further suggestions or would like clarification on any part of our response, please feel free to let us know. We would be very happy to continue the discussion.
>
> We truly appreciate the time and effort you have devoted to reviewing our work.
>
> Best regards,
>
> The Authors

---

### Official Review · Reviewer_3q7W · 2025-11-01

**Soundness:** 3
**Presentation:** 3
**Contribution:** 2
**Rating:** 4
**Confidence:** 4

**Summary:**

This work introduces Critical Token Fine-tuning (CFT), a model-free approach to identifying critical tokens in LLM reasoning tasks. Compared with other token-level importance methods, CFT uses counterfactual substitution with only one or two greedy rollouts per token, enabling efficient evaluation without auxiliary models.

**Strengths:**

- CFT offers a practical and efficient way to identify critical tokens without training auxiliary models.
- The paper is clearly written with rigorous notation.
- The experiments are detailed and compare across many LLM base models and datasets.
- The authors focus on practical usage and include RL result comparisons after CFT, aligning with trends in LLM post-training pipelines.

**Weaknesses:**

1. The paper does not present the loss differences between critical and non-critical tokens. Critical tokens may naturally have higher uncertainty and therefore require more focus during tuning.

2. All models are trained for three epochs without a validation set or early stopping, raising a major concern: if critical tokens are naturally underlearned, CFT may obviously surpass other methods under this configuration. In real scenarios, models are typically trained to convergence (and often use a validation set to avoid overfitting), making the final result uncertain; CFT’s advantage may be primarily faster convergence rather than better asymptotic performance.

3. The problem has already been studied; some related works are missing (e.g., [1]), and more baselines should be included for comparison (e.g., Rho-1 [2]).

4. It is better to show which tokens are identified as critical, and to demonstrate that these identified tokens indeed correspond to truly critical ones (for example, by comparing them with those recognized by human experts, not necessarily to be the same, but need to be explained).

5. The counterfactual idea appears effective but not surprising.

[1] Disentangling Reasoning Tokens and Boilerplate Tokens For Language Model Fine-tuning. ACL 2025 Findings.
[2] Not all tokens are what you need for pretraining. NeurIPS 2024.

**Questions:**

- Can you report the loss comparison between critical tokens and non-critical tokens during the training process? For example, do critical tokens naturally have higher uncertainty and thus require more focus while tuning?

- Would you provide a fairer comparison using the best validation epoch for each method (CFT and SFT, DPO)? This can clarify whether CFT’s advantage is primarily faster convergence rather than better final performance.  e counterfactual procedure.

---

> ### Author Response · Authors · 2025-11-18
> **Rebuttal 1**
>
> ## Weaknesses
> **1. The paper does not present the loss differences between critical and non-critical tokens. Critical tokens may naturally have higher uncertainty and therefore require more focus during tuning.**
>
> **Response 1:**
>
> We compared the loss dynamics of **critical vs. non-critical tokens** during SFT and CFT using Qwen2.5-7B. For CFT, non-critical token loss is logged but not optimized.
>
> **Loss Comparison Between Critical and Non-Critical Tokens**
>
> | Method | Epoch | Critical Loss | Non-Critical Loss |
> |-|-|-|-|
> | **SFT** | 0.0 | 0.369 | 0.401 |
> |        | 0.5 | 0.0434 | 0.0503 |
> |        | 1.0 | 0.0042 | 0.0661 |
> |        | 1.5 | 0.000146 | 0.0254 |
> |        | 2.0 | 0.000063 | 0.00745 |
> |        | 2.5 | 0.0000007 | 0.00578 |
> |        | 3.0 | 0.000959 | **0.00429** |
> | **CFT** | 0.0 | 0.369 | 0.401 |
> |        | 0.5 | 0.0617 | 0.107 |
> |        | 1.0 | 0.00339 | 0.130 |
> |        | 1.5 | 0.000002 | 0.149 |
> |        | 2.0 | 0.000076 | 0.158 |
> |        | 2.5 | 0.000001 | 0.167 |
> |        | 3.0 | 0.000171 | **0.154** |
>
>
> 1. **In SFT, critical-token loss decreases faster than non-critical-token loss**, even though all tokens are trained equally and their initial losses are similar. This suggests that critical tokens naturally carry higher importance.
>
> 2. **In CFT, non-critical-token loss remains relatively high** (about 0.15 after training), while it becomes extremely small in SFT (around 0.004). This matches our expectation, since non-critical tokens are not optimized in CFT, which is consistent with Figure 3(a), where CFT models maintain substantially higher entropy than SFT during RL.
>
> This analysis confirms that critical tokens are inherently more important for learning, and that CFT emphasizes these tokens while preserving diversity on non-critical tokens.
>
>
>
>
> **2. All models are trained for three epochs without a validation set or early stopping, raising a major concern: if critical tokens are naturally underlearned, CFT may obviously surpass other methods under this configuration. In real scenarios, models are typically trained to convergence (and often use a validation set to avoid overfitting), making the final result uncertain; CFT’s advantage may be primarily faster convergence rather than better asymptotic performance.**
>
> **Response 2:**
>
> To verify whether the advantage of CFT is simply due to faster convergence under a fixed 3-epoch schedule, we conducted an additional control experiment with a held-out validation split. We removed 100 samples from the training set as a validation set and monitored **eval loss** throughout training (4 evaluations per epoch). For each model, we report the **best checkpoint** (lowest eval loss) and compare it with our CFT results.
>
> | Model | Best SFT (epoch) | AVG | GSM8K | Math | SVAMP | ASDiv | MAWPS | CARP | TabMWP | Minerva | Gaokao | Olympiad | College |
> | - | - | - | - | - | - | - | - | - | - | - | - | - | - |
> | Qwen2.5-3B (SFT) | 2.0 | 49.2 | 69.2 | 42.9 | 82.7 | 78.8 | 85.0 | 44.8 | 47.3 | 15.4 | 37.7 | 17.5 | 20.4 |
> | Qwen2.5-3B (CFT) | – | 55.1 | 71.9 | 52.5 | 81.2 | 83.1 | 90.9 | 53.6 | 54.6 | 20.2 | 45.2 | 19.9 | 33.3 |
> | Qwen2.5-7B (SFT) | 0.25 | 53.8 | 75.3 | 48.9 | 86.5 | 80.9 | 88.4 | 48.9 | 55.1 | 21.0 | 41.8 | 25.2 | 20.2 |
> | Qwen2.5-7B (CFT) | – | 59.2 | 81.5 | 59.8 | 90.2 | 85.8 | 92.6 | 53.9 | 53.1 | 26.8 | 48.6 | 28.4 | 30.1 |
> | Qwen3-8B (SFT) | 2.75 | 60.9 | 83.1 | 68.6 | 91.1 | 84.5 | 91.0 | 54.1 | 44.9 | 31.6 | 56.1 | 36.9 | 28.3 |
> | Qwen3-8B (CFT) | – | 63.5 | 88.0 | 70.7 | 93.5 | 87.9 | 93.8 | 56.1 | 52.6 | 32.7 | 58.2 | 37.2 | 28.0 |
> | Llama3.1-8B (SFT) | 2.0 | 38.3 | 56.2 | 18.9 | 71.1 | 73.2 | 90.4 | 23.3 | 48.3 | 6.2 | 20.5 | 3.7 | 9.2 |
> | Llama3.1-8B (CFT) | – | 39.2 | 55.6 | 18.9 | 70.1 | 72.7 | 87.4 | 21.9 | 57.2 | 8.5 | 19.7 | 4.7 | 14.1 |
> | OLMo2-7B (SFT) | 1.0 | 40.3 | 68.6 | 19.7 | 77.9 | 73.4 | 88.4 | 25.9 | 49.3 | 4.4 | 20.5 | 3.9 | 11.5 |
> | OLMo2-7B (CFT) | – | 42.3 | 74.3 | 21.0 | 75.2 | 78.7 | 92.0 | 20.3 | 61.2 | 4.8 | 21.6 | 4.3 | 12.1 |
>
>
> - Most SFT models reach their best validation checkpoint within 0.25–2 epochs, showing that a 3-epoch schedule is not underfitting.
> - Even with early stopping and best validation-selected checkpoints, **SFT remains consistently worse than CFT** across all models and benchmarks.
> - Therefore, the performance gains stem from **better token-level supervision**, not from faster convergence.
>
> These results confirm that CFT achieves **better asymptotic performance**, not merely quicker convergence.

---

> > ### Author Response · Authors · 2025-11-18
> > **Rebuttal 2**
> >
> > ## Weaknesses
> >
> > **3. The problem has already been studied; some related works are missing (e.g., [1]), and more baselines should be included for comparison (e.g., Rho-1 [2]).**
> >
> > **Response 3:**
> >
> > We will add [1] to the Related Work in the revised version and have included **Rho-1** as an additional baseline. Rho-1 requires comparing the target model with a fine-tuned reference model. For each base model, we follow the original design and use the full fine-tuned checkpoint (from Table 1) as its reference. The full comparison table is shown below.
> >
> > | Model | Method | Avg | GSM8K | Math | SVAMP | ASDiv | MAWPS | CARP | TabMWP | Minerva | Gaokao | Olympiad | College |
> > | - | - | - | - | - | - | - | - | - | - | - | - | - | - |
> > | Qwen2.5-3B | Rho-1 | 49.8 | 69.1 | 43.2 | 82.9 | 78.9 | 84.9 | 46.4 | 49.4 | 15.8 | 38.4 | 17.5 | 21.2 |
> > | Qwen2.5-3B | Ours | 55.1 | 71.9 | 52.5 | 81.2 | 83.1 | 90.9 | 53.6 | 54.6 | 20.2 | 45.2 | 19.9 | 33.3 |
> > | Qwen2.5-7B | Rho-1 | 52.6 | 77.3 | 47.8 | 88.7 | 81.9 | 89.8 | 47.1 | 46.9 | 20.6 | 41.0 | 22.5 | 15.0 |
> > | Qwen2.5-7B | Ours | 59.2 | 81.5 | 59.8 | 90.2 | 85.8 | 92.6 | 53.9 | 53.1 | 26.8 | 48.6 | 28.4 | 30.1 |
> > | Qwen3-8B | Rho-1 | 60.7 | 83.2 | 68.9 | 91.5 | 83.9 | 91.3 | 53.4 | 45.5 | 31.6 | 55.1 | 35.7 | 27.8 |
> > | Qwen3-8B | Ours | 63.5 | 88.0 | 70.7 | 93.5 | 87.9 | 93.8 | 56.1 | 52.6 | 32.7 | 58.2 | 37.2 | 28.0 |
> > | Llama3.1-8B | Rho-1 | 37.8 | 56.6 | 18.1 | 65.9 | 72.7 | 90.2 | 22.6 | 47.6 | 9.2 | 18.2 | 4.3 | 10.1 |
> > | Llama3.1-8B | Ours | 39.2 | 55.6 | 18.9 | 70.1 | 72.7 | 87.4 | 21.9 | 57.2 | 8.5 | 19.7 | 4.7 | 14.1 |
> > | OLMo2-7B | Rho-1 | 40.6 | 69.2 | 19.5 | 76.8 | 75.6 | 91.7 | 21.9 | 46.5 | 7.7 | 21.3 | 4.3 | 12.4 |
> > | OLMo2-7B | Ours | 42.3 | 74.3 | 21.0 | 75.2 | 78.7 | 92.0 | 20.3 | 61.2 | 4.8 | 21.6 | 4.3 | 12.1 |
> >
> >
> > Across all backbones, our CFT method consistently achieves higher accuracy than Rho-1, showing that counterfactual-based critical token identification aligns better with reasoning correctness. We will include the full table and discussion in the revised version.
> >
> >
> > **4. It is better to show which tokens are identified as critical, and to demonstrate that these identified tokens indeed correspond to truly critical ones (for example, by comparing them with those recognized by human experts, not necessarily to be the same, but need to be explained).**
> >
> > **Response 4:**
> >
> > 1. **Token distribution analysis.** Using Qwen2.5-7B as an example, the most frequent *critical tokens* are numerical tokens (`0, 2, 1, 5, 4`), while the most frequent *non-critical tokens* are common function words or punctuation (`' '`, `the`, `of`, `,`, `=`). This aligns with intuition: numerical values directly determine correctness in mathematical reasoning.
> >
> > 2. **Model-wide consistency.**  Appendix Figure 5 shows that, across all models, critical tokens are predominantly numbers. In Appendix C.3 (Figure 6), we further compare:
> >    (a) training only on numeric tokens, and
> >    (b) training only on numeric tokens within the critical-token set.
> >    Both variants underperform the full CFT mask, indicating that **a small number of non-numeric critical tokens (e.g., operators or logical connectors) are also essential**. This supports the functional necessity of the identified critical tokens.
> >
> > 3. **Release of token-level annotations.**  In the *Reproducibility Statement*, we have included an anonymous link to our code release. It contains the full token-level criticality annotations for every model. Each token is labeled with a pair `[a, b]`, where `a` and `b` indicate whether replacing it with the top-2 or top-3 alternative preserves correctness:
> >    - `[0,0]` → critical (both replacements fail)
> >    - `[1,0]`, `[0,1]`, `[1,1]` → non-critical
> >    This enables reviewers to directly inspect and verify which tokens are labeled critical.
> >
> > Together, these analyses and released annotations demonstrate that our method identifies tokens that are intuitively and functionally crucial for correct reasoning, with full transparency and reproducibility.

---

> > > ### Author Response · Authors · 2025-11-18
> > > **Rebuttal 3**
> > >
> > > ## Weaknesses
> > >
> > > **5. The counterfactual idea appears effective but not surprising.**
> > >
> > > **Response 5:**
> > >
> > > While the counterfactual idea itself is simple, our work reveals a surprising use of it in the **SFT stage**, which differs from prior approaches in both design and outcome:
> > >
> > > 1. **Distinct from existing token-selection methods.** Prior methods, such as Rho-1 (pre-training) and TIS-DPO / cDPO (DPO stage), require additional auxiliary models. SHAD+RFT and DFT reweight tokens but still **train all tokens** in the response.
> > >    In contrast, our method performs token selection directly during SFT, requires **no extra models**, and **updates only the truly indispensable tokens**.
> > >
> > > 2. **Only ~12% of tokens are needed.** We show that updating **fewer than 12%** of response tokens consistently outperforms full SFT across all backbones—demonstrating that genuine token sparsity (not just reweighting) is both feasible and highly effective.
> > >
> > > 3. **Better diversity and RL initialization.** CFT-trained models maintain **higher entropy** during RL and yield superior final performance, aligning with intuition: training every token enforces strict imitation, while training only critical tokens avoids collapse and provides a stronger RL starting point.
> > >
> > > Overall, even if counterfactual substitution is conceptually straightforward, our results show that its **SFT-stage, model-free, sparse-token formulation** leads to unexpectedly strong improvements that are not achieved by prior methods.
> > >
> > > ## Questions
> > >
> > > **1. Can you report the loss comparison between critical tokens and non-critical tokens during the training process? For example, do critical tokens naturally have higher uncertainty and thus require more focus while tuning?**
> > >
> > > **Response 1:**
> > >
> > > Yes. We performed this analysis and provide the full results in **Weakness 1 — Response 1**.
> > >
> > > In summary, we compared the loss dynamics of critical and non-critical tokens during training using Qwen2.5-7B. We found that (1) even under standard SFT, critical-token loss decreases faster than non-critical-token loss, and (2) under CFT, non-critical-token loss remains substantially higher, which aligns with our entropy findings.
> > >
> > >
> > >
> > > **2. Would you provide a fairer comparison using the best validation epoch for each method (CFT and SFT, DPO)? This can clarify whether CFT’s advantage is primarily faster convergence rather than better final performance. e counterfactual procedure.**
> > >
> > > **Response 2:**
> > >
> > > We have conducted this analysis and report the results in **Weakness 2 — Response 2**. Specifically, we introduced a held-out validation set, monitored the eval loss throughout training, and selected the **best checkpoint** (early-stopped) for SFT. As shown in the results:
> > >
> > > - The best SFT checkpoints for SFT typically occur very early, indicating that 3 epochs are not underfitting.
> > > - More importantly, **even the best-validation SFT checkpoints remain consistently worse than CFT** across all models and all benchmarks.
> > > - This demonstrates that CFT’s gains are not due to faster convergence but reflect **better asymptotic performance** enabled by critical-token supervision.

---

> ### Author Response · Authors · 2025-11-26
>
> Dear Reviewer 3q7W,
>
> Wishing you a happy and blessed Thanksgiving!
>
> Thank you for your detailed and insightful review. We have added additional experiments and analyses to address all the questions you raised. If you have any further suggestions or would like clarification on any part of our response, please feel free to let us know. We would be very happy to continue the discussion.
>
> We truly appreciate the time and effort you have devoted to reviewing our work.
>
> Best regards,
>
> The Authors

---

### Official Review · Reviewer_33GQ · 2025-11-01

**Soundness:** 4
**Presentation:** 3
**Contribution:** 4
**Rating:** 8
**Confidence:** 3

**Summary:**

The authors propose Critical Token fine-tuning (CFT), a training method that only applies the loss to tokens that are regarded important. These tokens are identified through a counterfactual perturbation process - when replacing with other candidates all yields to incorrect prediction, it is chosen. The biggest merit of CFT is that it does not rely on compute-intensive rollouts when identifying such tokens.

**Strengths:**

1. The experimental set up is very well written - it was clear to tell why SFT, DFT, Entropy, Attn were chosen as baselines. Also, a consistent gain across 11 benchmarks and 5 base models very well support that CFT is an effective method.

2. The fact that "CFT-initialized checkpoints begin with higher entropy for RL training and that exploration is sustained" is a very good finding and could be adopted in future works. Specifically, there has been a lot of work showing that RL training improves Pass@1 performance at the cost of reducing entropy. With a better initialization, models could be trained with even more number of steps when CFT is adopted.

3. The ablation experiments are also well designed, especially, I enjoyed reading the findings in the experiment of Section 5.2 - applying CFT to identify critical tokens on another model family. This indicates that CFT could be applied in more general settings for response filtering.

**Weaknesses:**

I do not see any strong weaknesses in this paper.

**Questions:**

Can you make the font size of the figures bigger?

---

> ### Author Response · Authors · 2025-11-18
>
> Thank you for your positive assessment and recognition of our work. We appreciate your suggestion regarding figure readability. In the revised version, we will increase the font size of all figures to ensure clearer visualization.

---

### Author Response · Authors · 2025-12-01
**Summary**

**Dear AC, SAC, and PC**

We would like to extend our sincere gratitude to you for your recent efforts in improving the ICLR community and its review system. We are grateful that the reviewers found our method CFT to be **elegant and novel (4YV3, h8JF)**, praised its **effectiveness (33GQ, 3q7W, h8JF)**, and acknowledged the **thoroughness of our experiments and ablation studies (33GQ, 3q7W, h8JF)**. Below, we summarize the key points addressed in our rebuttal:

**1. Novelty and Conceptual Contribution of CFT**

(Addresses Reviewer 3q7W’s concern: “counterfactual idea appears effective but not surprising”; discussed in Response 5 for Reviewer 3q7W. Also **supported by positive novelty comments from 4YV3 and h8JF.**)

- CFT introduces a novel approach by selectively updating only those tokens critical to reasoning accuracy, identified through counterfactual perturbations, which does not require additional models.
- Unlike traditional supervised fine-tuning (SFT), which updates all tokens equally, or methods like Rho-1, TIS-DPO, cDPO, and DFT that either require extra models or still update all tokens, CFT focuses on the most crucial tokens, improving both model performance and exploration ability, while also enhancing reasoning ability and reinforcement learning initialization.

**2. Comparison with Rho-1**

(Addresses concerns from Reviewer 3q7W and Reviewer h8JF; answered in Response 3 for 3q7W and Response 1 for h8JF.)

- Reviewer 3q7W and Reviewer h8JF suggested a comparison with Rho-1, but it's important to note that Rho-1 is a pre-training critical token method that requires an additional model for token labeling.
- Nonetheless, we have addressed this request by adapting Rho-1's critical token identification to the SFT stage, conducting extensive comparisons across five models and eleven benchmarks.
- **Across all backbones, our CFT method consistently achieves higher accuracy than Rho-1**, demonstrating that counterfactual perturbation identifies decisive reasoning steps more reliably.

**3. Computational Efficiency of CFT**

(Addresses efficiency questions from Reviewer 4YV3 and Reviewer h8JF; answered in Response 1 for 4YV3 and Response 2 for h8JF.)

- We have implemented an efficient parallel critical identification strategy (Section 3.2) and proposed a lightweight two-stage variant to make CFT practical in both compute-rich and compute-limited scenarios. First, we annotate a small subset (~500 examples) using standard CFT to obtain $M_{\text{cft}}$. Then, we score tokens using the probability gap between the CFT model and the base model, avoiding full counterfactual rollouts for the entire dataset.
- This approach significantly reduces computation time: generating 500 annotated samples takes just 15.6 minutes on 8×A100 GPUs, while adding 1000 new annotations requires only 1.2 minutes on 1×A100 GPUs.
- The results confirm that efficient CFT maintains strong performance across five models and eleven benchmarks, offering a practical and computationally feasible solution for token-level supervision.

**4. Robustness to Random Seeds and Early-Stopping**

(Addresses Reviewer 3q7W’s concerns about early-stopping and Reviewer h8JF’s request for more seed experiments; covered in Response 2 for 3q7W and Response 3 for h8JF.)

- For Reviewer 3q7W, we performed additional validation-split experiments and evaluated multiple checkpoints per epoch. CFT consistently outperforms SFT even when SFT uses its **best validation checkpoint**, confirming the gains are not due to faster convergence.

- For Reviewer h8JF, we repeated all SFT/CFT experiments with multiple seeds and repeated RL experiments twice. The performance gains of CFT exceed the seed-level fluctuations, indicating that improvements are meaningful rather than stochastic.

----

We are committed to refining our paper to ensure clarity and accuracy. We have addressed all the comments and suggestions provided by the reviewers, and no new issues have been raised since we submitted our rebuttal on **November 18, 2025**. We greatly appreciate your time and effort in the review process.

**Sincerely,**
All authors

---

### Meta-Review · Area_Chair_vJLg · 2025-12-30

**Summary:**

Advantage:
1. The paper introduced a critical token fine-tuning approach by updating only those tokens important to reasoning accuracy, calculated with counterfactual perturbations, without additional models.
2. Experimental results showed the effectiveness of the proposed method.

Disadvantage:
1. The idea on critical tokens to be updated is not new. Although it did not require additional model, the computation cost on the counterfactual perturbations should be bigger.
2. The most important experiment compared to Rho-1 was missing in the original paper. Although added in the rebuttal, it was a pity that it did not have a chance to receive the acknowledgement for the effort. Besides, by checking the reproductive results of Rho-1, they were different than that in the original paper, however, the authors did not clarify the reason.

**Reviewer Concerns:**

Thanks for the rebuttal and the added experimental results.

Most reviewers had similar concerns on the novelty, and the convince of the experiments, except for Reviewer 33GQ.

However, I am not quite convinced by the rebuttal. I think neither do the most reviewers.

**Reviewer Scores:**

It received review scores 4, 4, 4, 8. The score 8 might be over-rated.

I do not think they will change the score, for no replies were received from the reviewers to clearly show that the score would be modified.

---

### Decision · Program_Chairs · 2026-01-26

Reject